

# Development of a diffuse reflectance probe for in situ measurement of inherent optical properties in sea ice

Christophe Perron[1,4], Christian Katlein[1,2], Simon Lambert-Girard[1], Edouard Leymarie[3], Louis-Philippe Guinard[1,4], Pierre Marquet[4,5], Marcel Babin[1]

[1]Takuvik Joint International Laboratory, Laval University (Canada)-CNRS (France), Québec city, G1V 0A6, Canada
[2]Alfred-Wegener-Institut Helmholtz-Zentrum für Polar- und Meeresforschung, Bremerhaven, 27570, Germany
[3]Laboratoire d'Océanographie de Villefranche-sur-Mer, Villefranche-sur-Mer, 06230, France
[4]CERVO Brain Research Centre, Laval University, Québec city, G1J 2G3, Canada
[5]Centre d'optique, photonique et laser, Laval University, Québec city, G1V 0A6, Canada

*Correspondence to*: Christophe Perron (christophe.perron@takuvik.ulaval.ca)

**Abstract.** Detailed characterization of the spatially and temporally varying inherent optical properties (IOPs) of sea ice is
necessary to better predict energy and mass balances, as well as ice-associated primary production. Here we present the development of an active optical probe to measure IOPs of a small volume of sea ice (dm$^3$) in situ and non-destructively. The probe is derived from the diffuse reflectance method used to measure the IOPs of human tissues. The instrument emits light into the ice by the use of optical fibre. Backscattered light is measured at multiple distances away from the source using several receiving fibres. Comparison to a Monte Carlo simulated lookup table allows to retrieve the absorption coefficient, the reduced
scattering coefficient and a phase function similarity parameter $\gamma$, introduced by Bevilacqua and Depeursinge (1999), depending on the two first moments of the Legendre polynomials, allowing to analyze the backscattered light not satisfying the diffusion regime. Monte Carlo simulations showed that the depth cumulating 95% of the signal is between 40±2 mm and 270±20 mm depending on the source-detector distance and on the ice scattering properties. The magnitude of the instrument validation error on the reduced scattering coefficient ranged from 0.07% for the most scattering medium to 35 % for the less
scattering medium over the two orders of magnitude we validated. Vertical profiles of the reduced scattering coefficient were obtained with decimeter resolution on first-year Arctic interior sea ice on Baffin Island in early spring 2019. We measured values of up to 7.1 m$^{-1}$ for the uppermost layer of interior ice and down to 0.15±0.05 m$^{-1}$ for the bottommost layer. These values are in the range of polar interior sea ice measurements published by other authors. The inversion of the reduced scattering coefficient at this scale was strongly dependent of $\gamma$, highlighting the need to define the higher moments of the
phase function. This novel developed probe provides a fast and reliable means for measurement of scattering into sea ice.



# 1 Introduction

The optical properties of sea ice are governing how incident shortwave radiation is partitioned into reflection, absorption and transmission at the surface of ice-covered polar oceans. Sea ice optical properties consequently have a significant influence on the climate and ecosystem of the polar regions. Anthropogenic global warming is lengthening the melt season (Markus et al., 2009), increasing dominance of first-year over multi-year ice (Comiso, 2012;Haas et al., 2008;Kwok et al., 2009;Maslanik et al., 2007;Nghiem et al., 2007) and reducing the thickness and area of the ocean covered by ice (Serrez et al 2007, Stroeve et al 2012). These transformations are enhancing heat deposition by incident shortwave radiation (Arndt and Nicolaus, 2014;Nicolaus et al., 2012;Perovich and Polashenski, 2012;Rösel and Kaleschke, 2012). These ice transformations also increase photosynthetically available radiation, which can result, in given conditions, to higher primary production in and under the ice (Arrigo et al., 2012;Arrigo et al., 2008;Fernández-Méndez et al., 2015). Over the past, the inherent optical properties (IOPs) of sea ice parameterized in climate models have been inverted from apparent optical properties (AOPs) measured above and below sea ice (Briegleb and Light, 2007;Holland et al., 2012;Katlein et al., 2020). However, measuring at top and bottom boundaries can't account for the strong depth dependency of the scattering properties inside sea ice. Comprehending this depth dependency is increasingly important to link sea ice morphological changes and light partitioning.

To assess IOPs vertical distribution, AOPs measured at the top and bottom boundaries have been coupled to IOPs estimations based on physical properties (Grenfell, 1983), diffuse attenuation of sunlight measured through a hole drilled in the ice (Ehn et al., 2008a;Ehn et al., 2008b;Light et al., 2008) or by the means of laboratory active optical measurements on core sections (Katlein et al., 2014;Light et al., 2015). These vertical measurements provide approximates to build a layered IOPs model representing sea ice, but the IOPs needs to be tuned based on assumptions in order to meet measured AOPs. A process which is time consuming and under-constrained. Active in situ measurement proved to be faster and convenient, because they are not coupled with another measurement method and are independent of sun enlightenment. But the analytical model used in the past to retrieve IOPs actively was based on the diffusion approximation (Maffione et al., 1998). This approximation holds for large optical paths such that it can't account for vertical heterogeneity of interior sea ice and is ineffective close to boundaries.

Investigation of the in situ IOPs of sea ice measured with an active source at smaller scale would provide constrained vertically resolved measurements which are time-efficient and convenient. Such a measurement would facilitate the study of in situ IOPs for the different layers, for different ice types, for different periods of the year and for different regions, feeding radiative transfer with more extensive and precise parameters for future climate models.

Estimating the radiative properties of sea ice based on its growth history is not yet possible. To reach this goal, the relation between structural and optical properties of sea ice needs to be better understood. Previous experiments have shown that IOPs of an interior sea ice lab sample can be correctly predicted based on the temperature and bulk salinity (Light et al., 2004). However, lab samples often undergo drastic physical changes when the brine drains out of the core during extraction and when



refrozen for conservation, altering the optical properties. Furthermore, the bottommost layer which shelters algae and the important surface scattering layer cannot be preserved in a lab. Relying on in situ small-scale observations of the IOPs rather than laboratory ones would help extend the structural-optical model to meet field data and to encompass every ice layer.

To study the temporal and spatial variations of the IOPs of sea ice in situ, we developed an active optical probe based on the principle of spatially resolved diffuse reflectance. The spatially resolved diffuse reflectance method is currently used in the biomedical field to characterize tissues of a concise and targeted volume during surgery without altering biological functions (e.g. Bargo et al., 2005;Kim et al., 2010;Rodriguez-Diaz et al., 2011;Schwarz et al., 2008;Thueler et al., 2003). In that case, calculated IOPs are linked to the biochemical and structural properties of human tissue (Bigio and Mourant, 1997;Brown et

al., 2009). Likewise, our probe measures IOPs for a small volume of ice (in the order of dm$^3$) at a precise location without altering the ice structure. Measuring small volumes allows obtaining vertically resolved IOPs profiles through the sea ice cover. The recorded vertical profiles of IOPs could serve directly to improve models of radiative transfer calculation or be linked to changes in the ice structure or the presence of biological activity. Spatially resolved diffuse reflectance is a relatively fast measurement method allowing obtaining IOPs readings in the field within minutes, making it easy to use for scientists.

Hence, this method could make the study of IOPs of sea ice more accessible and widespread.

The paper is separated as follows, we first present the theoretical background behind the spatially resolved diffuse reflectance method and we introduce the previous works on the IOPs of sea ice. Then, we present a validation of the method using reference optical media and an estimation of the depth of signal origin. Finally, we present in situ vertically resolved reduced scattering coefficients $b'$ in first-year Arctic interior sea ice. The $b'$ profiles were obtained close to Qikiqtarjuaq Island by the eastern

shore of Baffin Island in Canada between may 7$^{th}$ and may 10$^{th}$ 2019.

## 2 Background

### 2.1 Spatially resolved diffuse reflectance

Spatially resolved diffuse reflectance $R$ is the detected backscattered optical power at a distance $\rho$ away from an active source at the surface of a given medium normalized by emitted optical power. $R$ depends on the IOPs of the medium, on the source-

detector distance ρ, but also on other geometrical factors $G$. In our case, $G$ accounts for optical fibres core surface areas $A_{source}$ and $A_{det}$, optical fibres maximum acceptance angles $\Theta_{source}$ and $\Theta_{det}$ (linked to the numerical aperture $NA$ of the fibre). $R$ also depends on the refractive indices of the probed medium and of the overlaying environment $n_{med}$ and $n_{env}$.



## 2.2 Radiative transfer in sea ice

The fundamental IOPs involved in the radiative transfer equation are the absorption coefficient $a$, which describes the
probability of a photon being absorbed per unit of length, the scattering coefficient $b$, which describes the probability of a
photon being scattered per unit of length, and the phase function $p$, which describes the angular distribution of redirected
scattered photons (Mobley et al., 2010).

For highly scattering media, the phase function $p(\theta)$ can be expressed as a sum of Legendre polynomials $P_n$ using a limited
number of terms:

$$p_{Leg}(\theta) = \frac{1}{4\pi}\sum_{n=0}^{\infty}(2n+1)g_n P_n(\cos\theta),\tag{1}$$

where $g_n$ is the $n^{th}$ order moment of the phase function and $\theta$ denotes the angle between incident photon direction and photon
direction after scattering. Fewer moments can be used to describe the phase function in calculations as the number of scattering
events in the optical path augments. That is because the numerous and complex phase functions describing single interactions
along the optical path are smoothed when represented by one generalized phase function. Legendre polynomials are the
underlying basis of the simplified single-moment Henyey-Greenstein function broadly used for radiative transfer in sea ice
when satisfying the diffusion regime.

### 2.2.1 Diffusion regime

The diffusion regime stands if the detected photons have undergone a sufficiently large number of scattering events along their
path. This requirement is generally fulfilled if the magnitude of scattering is much greater than the magnitude of absorption
and if far from boundaries. This regime is the most commonly used for radiative calculation in sea ice. In the diffusion regime,
the detected power is only sensitive to the first-order moment $g_1$ (or simply $g$) of $p_{Leg}$, which corresponds to the average
cosine given by :

$$g_1 = g = 2\pi\int_{\theta=0}^{\pi}p(\theta)\sin\theta\cos\theta d\theta.\tag{2}$$

Photons scattered strictly forward or backward result in $g$ =1, -1 respectively. Photons scattered evenly over $\theta$, which is also
referred to as isotropic scattering, result in g =0.

The value of $g_1$ in sea ice depends strongly on the real refractive index of the brine channels, air bubbles and precipitated salts
inclusions relative to their surrounding environment (Light et al., 2004). Mobley et al. (1998), based on Mie theory calculation,
showed that $g_1$ of first-year interior sea ice ranges from 0.96 (very bubbly ice) to 0.99 (few bubbles) with a likely value of 0.98.





It is often assumed that drained ice has a $g_1$ closer to 0.86 because of the augmentation of drained channels relative refractive

index (e.g. Ehn et al., 2008a;Hamre et al., 2004;Light et al., 2004). Radiative transfer calculations sometimes assume a $g_1$ of

0.94 as an average for the whole vertical profile including drained and submerged sea ice (e.g. Light et al., 2008;Light et al.,

2015;Xu et al., 2012).

A specific case of the Legendre polynomials where $g_n = g_1^n$ allows expressing the phase function in a short form. This specific

case, called the Henyey-Greenstein phase function $p_{HG}(\theta)$ can be rewritten as:

$$p_{HG}(\theta) = \frac{1}{4\pi} \frac{1 - g_{HG}^2}{\left(1 + g_{HG}^2 - 2g_{HG}\cos\theta\right)^{3/2}}, \tag{3}$$

where, in that case, $g_{HG}$ is the same as $g_1$ or $g$. The Henyey-Greenstein phase function is the most commonly used in sea ice

radiative transfer models when the diffusion approximation is met. Grenfell and Hedrick (1983) demonstrate that the measured

phase function of laboratory-grown interior sea ice in single scattering regime is not well fitted by the single moment Henyey-

Greenstein function. Fitting $p_{HG}(\theta)$ on their measurements results in $g_{HG}$=0.59. We don't quite know if the single-moment

Henyey-Greenstein function is a good representation of the phase function of sea ice for in situ conditions.

The similarity principle states that for a homogeneous domain, far from boundaries and if the diffusion regime is obtained,

given the same $a$, any combination of $b$ and $g$ resulting in the same reduced scattering coefficient $b'$ results in the same

apparent optical properties (De Hulst and Christoffel, 1980):

$$b' = b(1 - g_1). \tag{4}$$

**2.2.2 Sub-diffusive regime**

The effect of $p(\theta)$ on the detected light can no longer be described by a single moment $g_1$ when a small number of collisions

occurred between source and detector. We then enter what we call the sub-diffusive regime. In that case, more moments need

to be included in $p(\theta)$ for precise calculation of apparent optical properties, including $R$.

Bevilacqua and Depeursinge (1999);Kienle et al. (2001) stress that, in a reflectance geometry, for $0.5 < \rho b' < 5$, the second-

order moment needs to be set free in Eq. (1) to correctly calculate R. Sea ice values found in literature for medium to high

scattering ice ($b'$=$10^1$ to$10^2$ m$^{-1}$), means that N=2 regime is met for $\rho$ in the order of few centimeters. Low scattering ice ($b'$=$10^{-1}$ to $10^0$ m$^{-1}$) for $\rho$ in the order of few centimeters results in a criterium below 0.5 and consequently falls into higher N regimes.

In order to limit the number of inverted parameters in our analysis to three, we assumed low scattering ice to also be in N=2

regime and dealt with the associated error.



A modified version of the Henyey-Greenstein phase function $p_{mHG}(\theta)$, introduced by Bevilacqua and Depeursinge (1999), allows setting its first two moments:

$$p_{mHG}(\theta) = \beta \frac{1}{4\pi} \frac{1-g_{HG}^2}{\left(1+g_{HG}^2 - 2g_{HG}cos\theta\right)^{3/2}} + (1-\beta)\frac{3}{4\pi} cos^2\theta, \tag{5}$$

where $\beta \in [0,1]$. The first term is the regular Henyey-Greenstein function fully characterized by its first moment $g_{HG}$. According to Eq. (6), the adjustment of $\beta$ and $g_{HG}$ allow for independent variation within a certain range of $g_1$ and $g_2$, the first two moments of the modified Henyey-Greenstein:

$$g_1 = \beta g_{HG}, \qquad g_2 = \beta g_{HG}^2 + \frac{2}{5}(1-\beta). \tag{6}$$

Trivially, the case $\beta = 1$ corresponds to the regular Henyey-Greenstein. Controlling $g_1$ and $g_2$ separately allows controlling two types of scattering: the anisotropic scattering by large particles compared to the wavelength (e.g. bubbles and brine channels) and the Rayleigh scattering by particles smaller than the wavelength (e.g. precipitated salts). Indeed, Rayleigh scattering is a quasi-isotropic process where $p(\theta)$ varies in $cos^2\theta$ only (van de Hulst, 1980).

The addition of a second free moment $g_2$ to describe $p(\theta)$ establishes a new similarity relation (Bevilacqua and Depeursinge, 1999;Wyman et al., 1989;van de Hulst, 1980), which has the main advantage to depend only on the phase function parameters:

$$\gamma = \frac{1-g_2}{1-g_1}. \tag{7}$$

Physically speaking, $\gamma$ indicates the weight of near-backward scattering in $p(\theta)$. Near-backward scattering is increasing as $\gamma$ is decreasing. For the scattering properties of sea ice and $\rho$ in the order of the few centimeters, we assume that $R$ is dependent on $a$, $b'$ and $\gamma$. Then, its analysis provides an estimation of these three parameters reflecting IOPs.

## 2.2 Previous works on the IOPs of sea ice

The highly scattering and impenetrable nature of sea ice makes IOPs difficult to deduce. Over the past, various techniques have been developed to estimate IOPs of sea ice which shall be summarized in the following.

### 2.2.1 Structural-optical theory

Grenfell (1983) described a theoretical framework to estimate the IOPs of sea ice from the distribution of size, shape and the refractive indices of gas bubbles, brine channels as well as precipitated salts included in sea ice. The total absorption coefficient can be formulated as the sum of the respective absorption coefficients $a$ of ice and inclusions weighted by their respective



volume fraction. Scattering properties were calculated assuming the inclusions to be collections of spheres. With that
assumption, Mie theory was used to retrieve the scattering coefficient $b$ and the phase function $p(\theta)$.

## 2.2.2 Cold laboratory measurements

Using structural-optical theory, Light et al. (2003a) and Light et al. (2004) determined the reduced scattering coefficient $b'$ of
an ice sample in a cold lab. They observed size and shape distributions of inclusions with a microscope. In parallel, Light et
al. (2003b) developed a Monte Carlo inverse model to retrieve IOPs from active optical observations of an ice sample. Reduced
scattering coefficients estimated from a theoretical framework and from active optical observation were compared for different
temperatures and salinities. The comparison was used to adjust the theoretical framework for the contribution of certain
processes. The estimation of the volume of gas, the brine channels drainage during the measurement procedure, the scattering
by hydrohalite salts, the brine channels merging and air bubbles merging and escaping were adjusted for in the model.

Light et al. (2015) measured the scattering coefficient $b$ of cylindrical natural sea ice core samples cut in sections. The 10 cm
diameter and 10 cm long sections were introduced in a cylindrical chamber. A Tungsten-Halogen lamp followed by diffuser
plate and an aperture emitted multispectral light incidentally on the centre of the samples upper surface. Transmitted light was
measured at the bottom by the means of an optical fibre coupled to a spectrometer. Comparison between measured
transmittance and 2D Monte Carlo simulated transmittance allowed to retrieve $b'$. $b$ value was retrieved assuming a $g_1$ of 0.94
(see Eq. (4)).

Few authors built refrigerated tanks reproducing sea ice growth conditions in order to take optical measurements. To our
knowledge, one author retrieved the IOPs of sea ice from it. Marks et al. (2017) retrieved the IOPs by comparing albedo α
and diffuse attenuation $k$ measurements to the output of a DISORT simulation (Stamnes et al., 1988). IOPs input in the
DISORT model were tuned for simulations to fit measurements.

Grenfell and Hedrick (1983) measured $p(\theta)$ of laboratory-grown sea ice using a goniometer. Their sample was thinner than
the scattering mean free path assuring them to respect the single scattering regime.

## 2.2.3 In situ measurements

In situ estimations were based either on passive AOPs observations or active optical measurements. A variety of authors
inferred IOPs of sea ice optical layers using transmittance $T$, α and/or an estimation of $k$ in the ice (e.g. Ehn et al.,
2008a;Hamre et al., 2004;Light et al., 2008;Light et al., 2015;Mobley, 1998;Xu et al., 2012). Measurements were compared
to simulated values obtained from a radiative transfer model (e.g. DISORT, 4DOM, Hydrolight, AccuRT) (Grenfell,
1991;Mobley et al., 1993;Stamnes et al., 1988). In the model, first guesses of optical properties of individual layers were based





on structural observations and, then, the IOPs were subsequently adjusted for the model to match observations. Light et al. (2015) also used laboratory measurements to constrain first guesses of interior ice IOPs.

IOPs of interior sea ice were also estimated using an active light source. Maffione et al. (1998) observed the beam spread function with a rotating collimated laser diode emitting sideward and a detector placed 15 to 50 cm apart horizontally. The beam spread function and the diffusion theory helped to retrieve average IOPs values for interior sea ice. Trodahl et al. (1987) observed the spatially resolved intensity of monochromatic light backscattered at the surface of the ice and under the ice. Fitting these measurements to Monte Carlo simulations, they retrieved the averaged $b'$ for interior sea ice. To our knowledge, vertically resolved active measurements of the IOPs of sea ice in situ have not been attempted yet.

## 3 Methods

The scientific objective behind the development of the probe is to document in situ inherent optical properties (IOPs) of sea ice. To do so, we developed a probe based on the spatially resolved diffuse reflectance technique. Conceptually, the instrument emits light into the ice by means of an optical fibre. Backscattered light is measured at multiple distances $\rho_i$ away from the source at the medium interface using other fibres. The measured reflectance $R_{mes}(\rho_i)$ is compared to reflectance derived from Monte Carlo simulations mimicking the configuration of the experimental setup. A precomputed lookup table and an inverse algorithm allow calculating $a$, $b'$ and $\gamma$ of a small, concise volume in the order of the dm$^3$ corresponding to the region probed by the detected light. During field tests, we fixed $a$ and $\gamma$ values and obtained vertically resolved profiles of the dominant and easier to retrieve b$'$ in interior sea ice.

### 3.1 Experimental setup

Figure 1 shows a schematic of the experimental setup. The 2" diameter probe head 3D-printed in polycarbonate (with 3 extended, Ultimaker™, Utrecht, Netherlands) was designed to accurately fit an auger hole drilled through the ice. The location of the fibres allowed to measure IOPs sideward from the edge of the hole. The wall of the auger hole was smooth enough for all fibres to practically touch the ice surface (~ 1 mm interstice). Monte Carlo simulations demonstrated that an interstice of 1 mm results in an underestimation smaller than 5% on $R$.

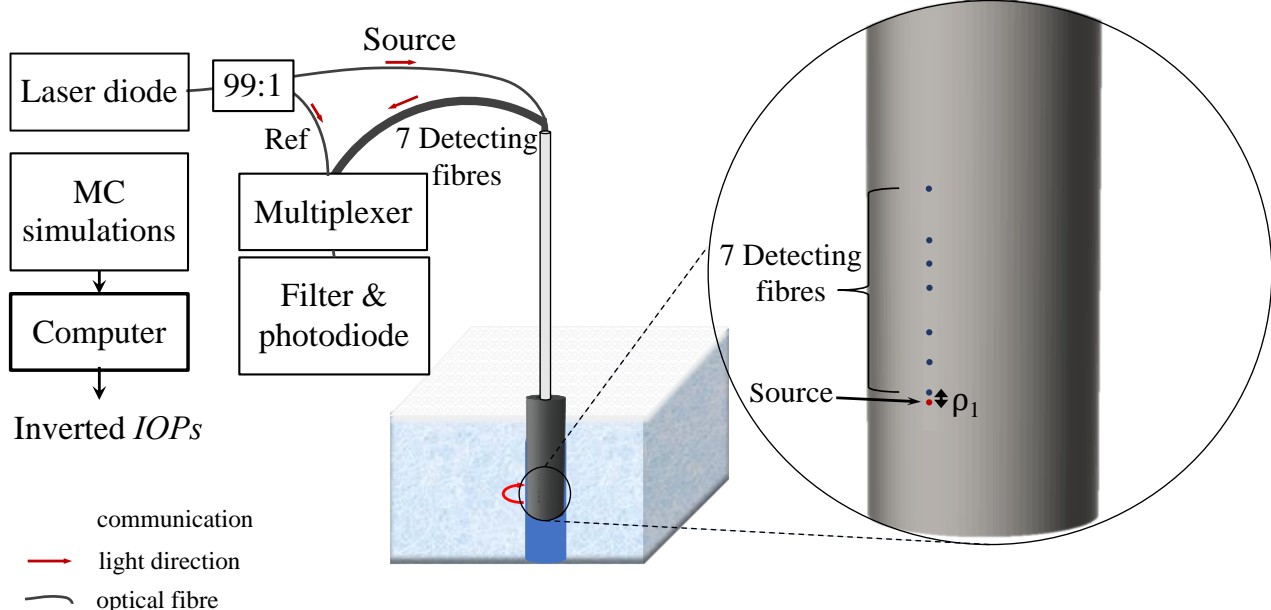

**Figure 1: Experimental setup schematic. The two-inch probe was designed to fit an auger hole drilled through the ice and to measure sideward on the edge of the hole. A laser diode emitted at $\lambda$=633 ± 2 nm with full-width half-maximum of 1.4 nm and optical power of 300 mW. A 99:1 fibre optic coupler divided optical power between a leg used to guide light up to the probe head where injection into sea ice occured and the reference leg. Detecting optical fibres at distances $\rho_{1-7}$ collected backscattered laser light (curved red arrows). An optical multiplexer selected the fibre to be read by a photodiode. A reflective bandpass filter centred at $\lambda$=633 nm with full-width half-maximum of 5 nm was placed before the photodiode to reject sunlight. A single-board computer with an easy-to-operate touch screen controlled the multiplexer, obtained $\Phi$ readings from the photodiode and did a field inversion on $b'$.**

The light source was a laser diode (PSU-III-DEL, Changchun New Industries[TM], Changchun, China) emitting a spectrum

centered at a wavelength $\lambda$ =633 ± 2 nm with 1.4 nm full-width half-maximum. The optical power of the laser was up to 300

mW at the tip of the emitting fibre with variations of less than 1% after a warm-up of five minutes. A 99:1 fibre optic coupler

(TM200R1S1A, Thorlabs[TM], Newton, United States) splitted optical power between the reference leg (~1% of power) and a

leg used to guide light up to the probe head where injection into sea ice occurred (~ 99 % of power). Seven optical fibres were

positioned to collect backscattered laser light at source-detector distances $\rho_{1...7}$ of 2, 8, 14, 23, 28, 33 and 43 mm at the ice

interface (see Figure 2 a). The printing allowed a precision of ±20 microns on fibres position. Source and detecting silica fibres

(FT400UMT, Thorlabs[TM], Newton, United States) had a diameter of 400 microns and a *NA* of 0.41 at $\lambda$ =633 nm (or

$\Theta_{det}$ = 18.3 ° in ice). The measured $\Theta_{source}$ for the combination of laser, fibre optic coupler and source fibre was 7.3 ° in ice.

The source reference fibre and the detecting fibres were connected to an optical multiplexer (MPM-2000, Ocean Insight[TM],



formerly Ocean Optics, Dunedin, United States) which selected the fibre to be measured. The output of the multiplexer was
connected by the means of the same type of optical fibre to a photodiode (PH100-Si-HA-D0, Gentec-EO™, Quebec City,
Canada). All detecting fibres, source reference fibre and a dark measurement were read by the photodiode in less than 30
seconds. A reflective bandpass filter centred at $\lambda$ =633 nm with full-width half-max of 5 nm was placed before the photodiode
to reject sunlight. A single-board computer with a touch screen (Lattepanda DFR0444, DFRobot ™, Shanghai, China)
controlled the multiplexer and recorded the photodiode radiant flux $\Phi$ measurements. The touch screen and touch pencil
allowed to control the instrument without taking out gloves, making operations under cold conditions more convenient.

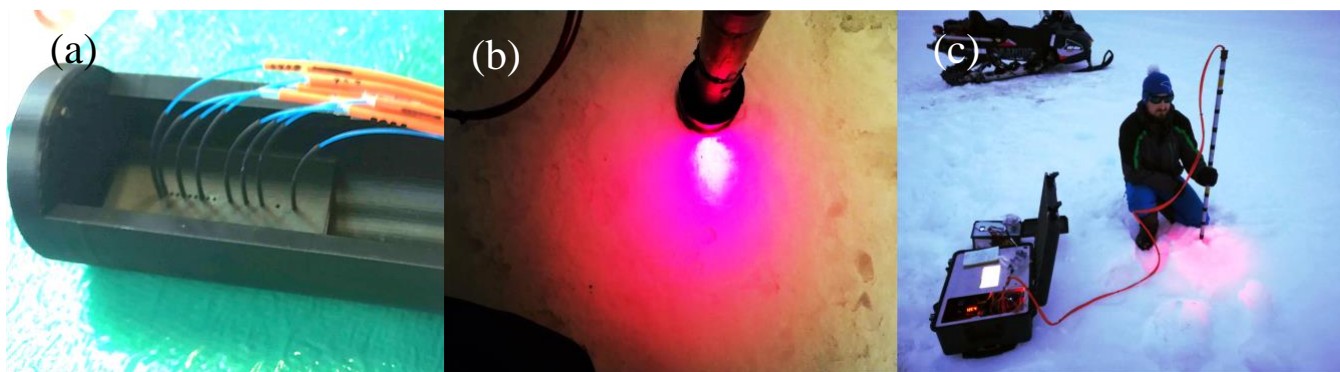

**Figure 2: (a) Probe head interior. Source and detecting fibres were held in place with heat shrink tubes and glue. Fibres bend radii are meant to respect the minimum long-term limit of 40 mm. (b) Probe head inserted in a 2" auger hole drilled in sea ice. (c) Probe operated on sea ice close to Qikiqtarjuaq Island on the coast of Baffin Bay.**

The measured reflectance $R_{mes}$  was calculated following:

$$R_{mes,i} = c_i \cdot \frac{\Phi_i - \Phi_{bg,i}}{\eta \cdot \Phi_{ref}},  \tag{8}$$

where c is the calibration factor that accounts for the optical power losses through the system and the mismatch with Monte
Carlo simulations. $\Phi_i$ is the backscattered radiant flux detected at the surface of the medium by fibre $i$, $\Phi_{bg}$ is the sum of the
sunlight background radiant flux and dark noise, $\Phi_{ref}$ is the radiant flux detected by the source reference fibre and $\eta$ is the
coupler split ratio $\Phi_{source}/\Phi_{ref}$.

## 3.2 Monte Carlo simulations for generation of the lookup table

Because we measure in the sub-diffusive regime, we cannot rely on an analytical solution to retrieve IOPs. Instead, we rely on
a Monte Carlo numerical approach. We simulated the spatially resolved diffuse reflectance $R_{sim}(\rho, a, b', \gamma)$ using the Monte
Carlo software SimulO (Leymarie, 2010). The software allows creating 3D environments with complex shapes, sources and





detectors. It has been used and validated multiple times for research in ocean optics (e.g. Babin et al., 2012;Leymarie et al., 2010;Massicotte et al., 2018).

Figure 3 illustrates the numerical environment designed to simulate $R_{sim}(\rho, a, b', \gamma)$ for our geometry. Light was emitted from the tip of the source fibre toward probed medium. Photons were emitted in a direction inside $\Theta_{source} = 7.26°$ in ice. The emission angular profile of photons followed a Lambertian distribution. The medium representing sea ice was given a refractive

index $n_{med}$, an absorption coefficient $a$, a scattering coefficient b and a phase function $p(\theta)$. We used the modified Henyey-Greenstein phase function $p_{mHG}(\theta)$ (see Eq. (5)). Thus, $g_1 = \beta g_{HG}$ and $g_2 = \beta g_{HG}^2 + \frac{2}{5}(1 - \beta)$ are defined accordingly. The environment overlaying the medium had a refractive index $n_{env}$. Two variations of the numerical environment were implemented. For measurement inside sea ice, the probed medium had the refractive index of ice ($n_{med}= 1.31$) and the overlaying environment had the refractive index of water ($n_{env}= 1.33$). For calibration and validation with solutions of

microspheres, $n_{med}= 1.33$ and $n_{env}=1$.

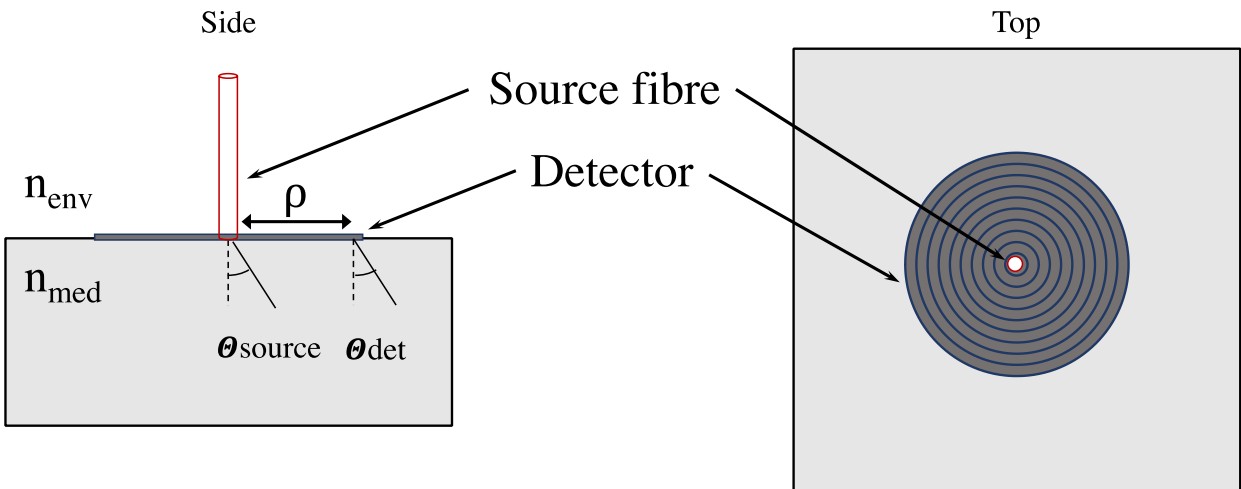

**Figure 3: Schematic of the numerical environment used to simulate $R_{sim}(\rho, a, b', \gamma)$ with the Monte Carlo method.**

Detecting fibres were replaced by a circular detector to collect photons over a larger area and, thereof, to reduce calculation time. It counted photons crossing with an incident half angle $\leq \Theta_{det} = 18.25°$ in ice ($NA=0.41$). Photon counts were

azimuthally averaged for 10 circular bins evenly distributed along the radius ($\rho$=0, 6.7, 13.3, 20.0, 26.7, 33.3, 40.0, 46.7, 53.3 and 60 mm) in order to cover all the surface. The replacement of detecting fibres by a detector induced an error of less than 0.4 % on reflectance. The error came from Fresnel reflection on the tip of the detecting fibres which was not accounted for (Hecht and Zajac, 1974).



As explained in Sect. 2.2.2, inverse procedure provides parameters $a$, $b' = b(1 - g_1)$ and $\gamma = \frac{1-g_2}{1-g_1}$. Thus, to facilitate a, b'

and $\gamma$ determination, we fixed $g_1$=0.98, a representative value for interior sea ice in situ (Mobley et al., 1998). We ran the

simulations for 20 values of $a$ from 0.01 m$^{-1}$ to 3 m$^{-1}$, 92 values of $b'$ from 0.05 m$^{-1}$ to 300 m$^{-1}$ and 7 values of $\gamma$ from 0.8 to

1.98. Absorption and scattering properties were selected to cover sea ice IOPs as known from previous studies (Ehn et al.,

2008a;Light et al., 2008;Light et al., 2015;Mobley et al., 1998;Trodahl et al., 1987). The range of $\gamma$ was limited by the

mathematical condition on $p_{mHG}(\theta)$ where $\beta \in [0, 1]$ (see Eq. 5) (Bevilacqua and Depeursinge, 1999).

$R_{sim}(\rho, a, b', \gamma)$ was simulated for every combination of $a$, $b'$ and $\gamma$ with the numerical environment shown on Figure 3 . For

every IOPs combination, 10 simulations of 10$^6$ photons were computed and the normalized standard deviation $\sigma(R_{sim})/R_{sim}$

between those simulations was obtained. $\sigma(R_{sim})/R_{sim}$ was below 5% when $b' \geq 2$ m$^{-1}$ and was always less than 12%. The

large volume of calculation (20x92x7x10 simulations of 10$^6$ photons) required the use of Compute Canada computation

resources. Even with high computation power, the 4D output matrix had an insufficient resolution to calculate precise IOPs.

To increase resolution, we interpolated $R_{sim}(\rho, a, b', \gamma)$ successively on $a$, $b'$ and $\gamma$ dimensions with linear regression. Also,

for the interpolated simulated spatially resolved diffuse reflectance $\bar{R}_{sim}(\rho, a, b', \gamma)$ to match detecting fibres positions $\rho_{1...7}$,

the matrix was linearly interpolated on the spatial dimension $\rho$. The final lookup table $\bar{R}_{sim}(\rho_{1...7}, a, b', \gamma)$ was a

7x250x395x200 matrix. For visualisation in the field, a lighter 7x1x395x1 matrix was implemented, fixing $a$ to 0.22 m$^{-1}$ and

$\gamma$ to 1.98.

**3.3 Inversion algorithm**

Retrieval of $a, b'$ and $\gamma$ was achieved by comparing $R_{mes}(\rho_{1-7})$ to every curve of an interpolated Monte Carlo simulated

lookup table $\bar{R}_{sim}(\rho_{1-7}, a, b', \gamma)$. The error $\chi^2$ for every variation of $a, b', \gamma$ was calculated following:

$$\chi^2(a, b', \gamma) = \sum_{i=1}^{7} \frac{\left(R_{mes}(\rho_i) - \bar{R}_{sim}(\rho_i, a, b', \gamma)\right)^2}{R_{mes}(\rho_i)} . \tag{9}$$

The simulation that fitted the best to the measurement was defined by the smallest element in the error matrix $\chi^2(a, b', \gamma)$. The

coordinates $a, b', \gamma$ of that element were the calculated IOPs (Figure 4). Many versions of Eq. (9) were tested for robustness.

Noticeably, the algorithm was less sensitive to noise when the subtraction was normalized by $R_{mes}(\rho_i)$ . Or said otherwise,

the algorithm was less sensitive when detecting fibres all had equal weights. The choice of a discrete interpolated matrix rather

than a continuous method like Levenberg-Marquardt was motivated by robustness. Comparing measurements to every discrete

element of $\bar{R}_{sim}(\rho_{1-7}, a, b', \gamma)$ insured we avoided incorrect inversion because of local minima in the error matrix. The

downsides were heavier calculation time and larger memory needs.



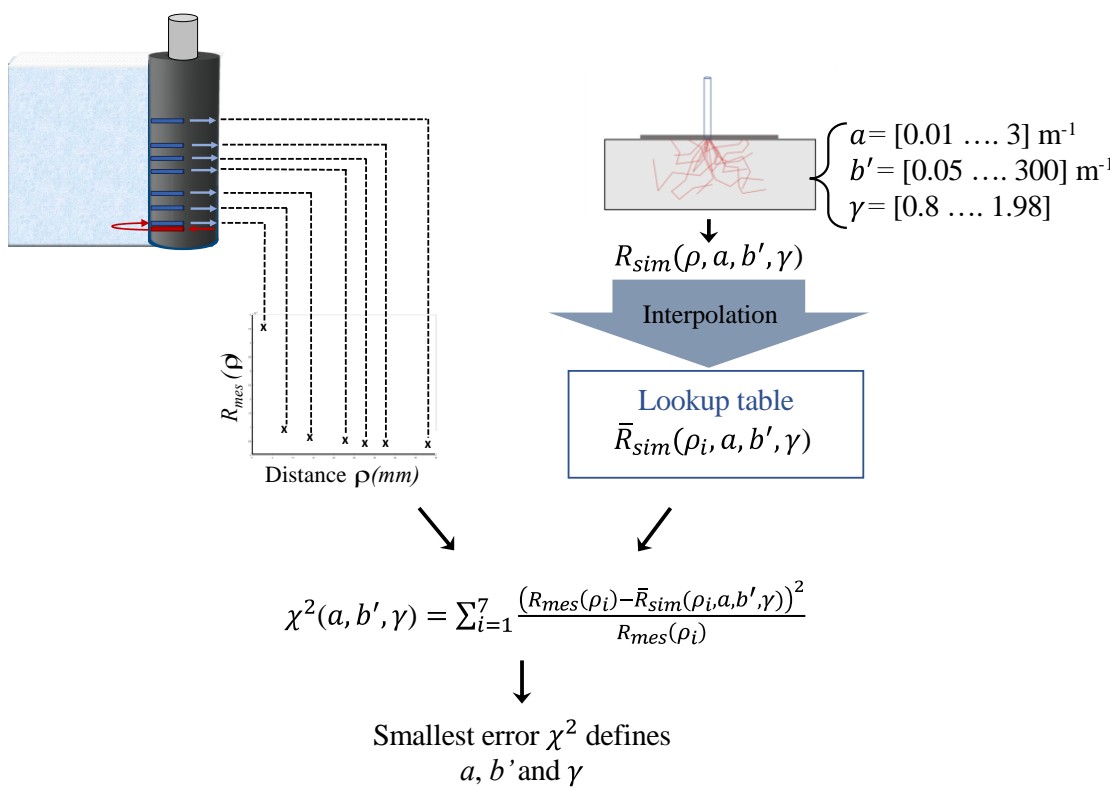

**Figure 4: Illustration of the inversion algorithm used to retrieve inherent optical properties of sea ice.**

## 3.4 Estimation of the depth of signal origin

The appropriate probed volume was evaluated based on two conditions. First, the signal of origin should reach deep enough
to average the contribution of a large number of inclusions. Inclusions causing scattering in young sea ice (brine channels, air
bubbles and precipitated salts) range from less than a micron up to rarely more than 10 mm (Light et al., 2003a;Perovich and
Gow, 1991). Second, the depth of signal origin should be small enough to resolve every optical layer when leaned horizontally.
The topmost and thinnest optical layer of sea ice (surface scattering layer) is typically no less than a couple of centimeters
(Ehn et al., 2008a;Light et al., 2008;Light et al., 2015). Based on these two criteria, we established that the ideal volume
measured by the probe shall be deeper and wider than roughly 10 mm to encompass a large number of inclusions and, in the
best scenario, shallower and narrower than 50 mm to resolve the optical properties of the surface scattering layer horizontally.

Bevilacqua (1998) demonstrated that depth of signal origin is roughly proportional to $\rho$ (for human tissues). Therefore, $\rho$ tunes
the probed volume. To verify the relation between $\rho$ and the depth of signal origin for sea ice, which scatters and absorbs



significantly less than human tissues, we ran Monte Carlo simulations. For this test, the numerical environment we used was
similar to the environment shown in Figure 3, except that a totally absorptive horizontal slab was inserted in the medium. The
slab was lowered from the surface downward by increments $\Delta z$ of 0.5 mm down to a depth $z$ of 30 mm and by increments $\Delta z$
of 4 mm deeper. The cumulative signal $R(z,\rho)/R_{max}(\rho)$ vs absorptive plate depth $z$ vs $\rho$ was evaluated for $a$=0.1m$^{-1}$, $b$=10
m$^{-1}$,100 m$^{-1}$ and 1000 m$^{-1}$ and g=0.94. We ran 40 simulations of 10$^6$ photons for every $z$ when $b$=10 m$^{-1}$ and 100 m$^{-1}$ and 10
simulations of 10$^6$ photons for every $z$ when $b$=1000 m$^{-1}$. The standard deviation between simulations was used to evaluate
the uncertainties. The depth, $z_{95}$ , were $R(z,\rho)/R_{max}(\rho)$ is 95 % was linearly interpolated on the $\rho$ dimension to obtain
estimation at fibre positions $\rho_{1\ldots7}$.

## 3.5 Calibration and validation using polystyrene microspheres in water

Prior to field tests, the probe was calibrated and IOPs measurements were validated using reference media. Our reference
media were suspensions of Polyscience™ polystyrene microspheres with a diameter of 1.93±0.01 μm in distilled water.
Changing the microspheres volume fraction allowed to tune $a$ and $b'$ of the medium. We obtained 4 reference points by a
series of dilutions (**Error! Reference source not found.**). The theoretical value of the absorption coefficient $a$ was calculated
by weighting water and polystyrene $a$ values (Kadhim, 2016) by their respective volume fraction. The theoretical values of
the $b'$ and $\gamma$ were calculated using Mie theory (Bohren et al., 1983). A magnetic stirrer ensured that the microspheres
concentrations were homogeneous during measurements. We did not measure higher concentrations, resulting in higher a and
$b'$, for technical reasons; The quantity of microspheres solution needed to meet the required solvent volume would have been
too great.

To calibrate our system, uncalibrated spatially resolved diffuse reflectance $R_{mes}^*(\rho_{1\ldots7})$ measured from dilution #1 was
compared to its closest corresponding simulation in $\bar{R}_{sim}(\rho_{1\ldots7}, a, b', \gamma)$. The calibration factor $c$ shown in Eq. (8) was
calculated following:

$$c_{1\ldots7} = \frac{\bar{R}_{sim}(\rho_{1\ldots7})}{R_{mes}^*(\rho_{1\ldots7})}.$$   (10)

The calibration factor accounted for the optical power losses through the system and the mismatch with simulations. The
calibration factor $c$ obtained on dilution #1 resulted in the lowest errors in the subsequent validation.




| Dilution order | Microspheres volume fraction •$10^{-3}$ [-] | Theoretical $a$ [m$^{-1}$] | Theoretical $b'$ [m$^{-1}$] | Theoretical $\gamma$ [-] | Theoretical $g_{Mie,1}$ [-] |
|---|---|---|---|---|---|
| 1* | 0.356 ± 0.001 | 1.47 ± 0.02 | 75 ± 1 | 1.951 ± 0.001 | 0.9205 ± 0.0005 |
| 2 | 0.1780 ± 0.0005 | 0.93 ± 0.02 | 37.5 ± 0.7 | 1.951 ± 0.001 | 0.9205 ± 0.0005 |
| 3 | 0.0356 ± 0.0001 | 0.507 ± 0.008 | 7.5 ± 0.1 | 1.951 ± 0.001 | 0.9205 ± 0.0005 |
| 4 | 0.01780 ± 0.00005 | 0.454 ± 0.007 | 3.75 ± 0.07 | 1.951 ± 0.001 | 0.9205 ± 0.0005 |

*Used for calibration

**Table 1: Theoretically calculated IOPs for 4 concentrations of polystyrene microspheres in water used to first calibrate and then validate the instrument measurements.**

For validation, calibrated $R_{mes}$ was entered in the inversion algorithm described in Sect. 3.3 Inversion algorithm  and $a$, $b'$ and $\gamma$  were retrieved at all 4 concentrations. Measurements were taken 10 consecutive times. This way, we retrieved the mean and the standard deviation on $a$, $b'$ and $\gamma$ . The means were compared to theoretically calculated values. The instrument validation error $e$ between measured IOPs and theoretically calculated IOPs is given by:

$$e = 100 \cdot \frac{(measurement - theoretical\ value)}{theoretical\ value}.$$ (11)

## 3.6 Field work

Using the spatially resolved diffuse reflectance method, we profiled first-year Arctic interior sea ice at 2 study sites around Qikiqtarjuaq Island next to Baffin Bay in Nunavut, Canada from May 7 to May 10, 2019 (Figure 5). One site was on snow-covered ice (67.59 N, 64.03 W) and one site was on bare ice (no snow accumulation) (67.49 N, 63.95 W). Air temperature was between -6°C and 3 °C and the sky was sunny with passing clouds for most of the sampling period. Both sites had a slightly positive freeboard. We were at the very beginning of the melt season and snow was starting to be slushy. A thin melt crust was present on snow at the snow-covered site. At the bottommost layer of the ice was a thin and pale algae layer. No other impurities were observed in the ice column.



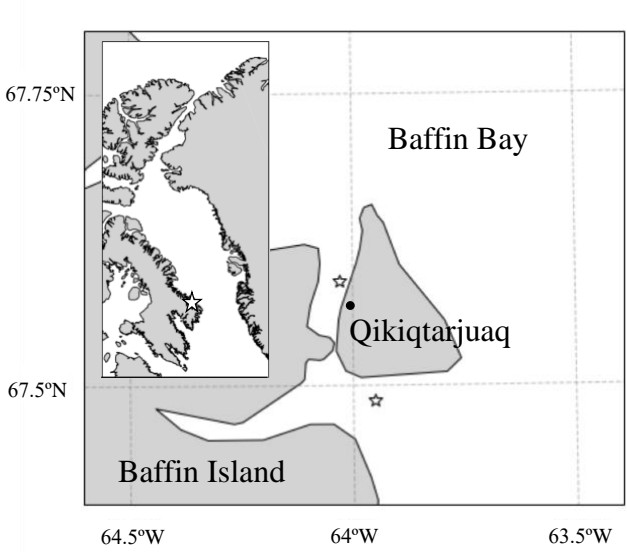

**Figure 5: Locations of the snow-covered and bare ice sampling sites visited on May 8th and 9th 2019 near Qikiqtarjuaq on the shore of the Baffin Bay in Nunavut, Canada.**

Sea ice thickness, snow thickness and freeboard were measured through the hole with a thickness gauge (Kovacs Entreprises TM,
Roseburg, United States).

After a warm-up of 5 minutes, the probe head was inserted inside a 2" auger hole (Figure 2 b-c). Emission reference flux $\Phi_{ref}$
and $\Phi_{1-7}$ were measured every 10 cm starting from the top and lowering the probe head until the bottom of the hole. When
the bottom was reached, the laser was shut down. Then, sunlight background $\Phi_{bg}$ was measured with the probe at every depth
on the way up. Field trials have shown that the sunlight background is significant as it can have the same order of magnitude
as the signal in the worst scenario (and is $10^4$ times smaller in the best scenario). For every depth, $R_{mes}(\rho_{1...7})$ was obtained
(see Eq. (8)) and $b'$ was inverted from it with fixed $a$ and $\gamma$. The output result is a profile of $b'$ vs depth in the ice.
Measurements were repeated in the same hole with a tent, with a tarpaulin covering the ground (at bare ice site only) and with
no cover to shield sunlight. The use of a tent or of a tarpaulin diminishes the sunlight background by roughly 1 and 2 orders
of magnitude, respectively.

After profiling, an ice core was retrieved next to the sampling site using an ice corer (Mark II 0.09 m diameter 1 m long corer,
Kovacs Entreprises TM, Roseburg, United States ). A picture of the ice core was obtained for qualitative observation of the ice
scattering properties. Ice temperature $T$ was measured at the centre of the core at 10 cm intervals using a high precision
thermometer (VWR International TM, Radnor, United States - ±0.1°C). For the measurement of bulk salinity $S$, the core was





cut in 10 cm sections. Ice sections were melted in plastic bags. The $S$ of the melted ice section was measured using a

conductometer (738-ISM, Mettler-Toledo InLab™, Colombus, United States).

# 4 Results

## 4.1 Depth of signal origin

Figure 6 shows $R(z,r)/R_{max}(\rho)$ vs absorptive plate depth $z$ at different $\rho$. The depth of signal origin was dependent on the scattering properties of the medium. When scattering was low ($b$=10 m$^{-1}$), as for interior ice, $z_{95}$ was 110±20 mm when

detecting at $\rho_2 = 8$ mm and was 270±20 mm when detecting at $\rho_6 = 43$ mm. When scattering was high ($b$=1000 m$^{-1}$), as for surface scattering ice, $z_{95}$ was 39±2mm when detecting at $\rho_2$ and was 78±4 mm when detecting at $\rho_6$.

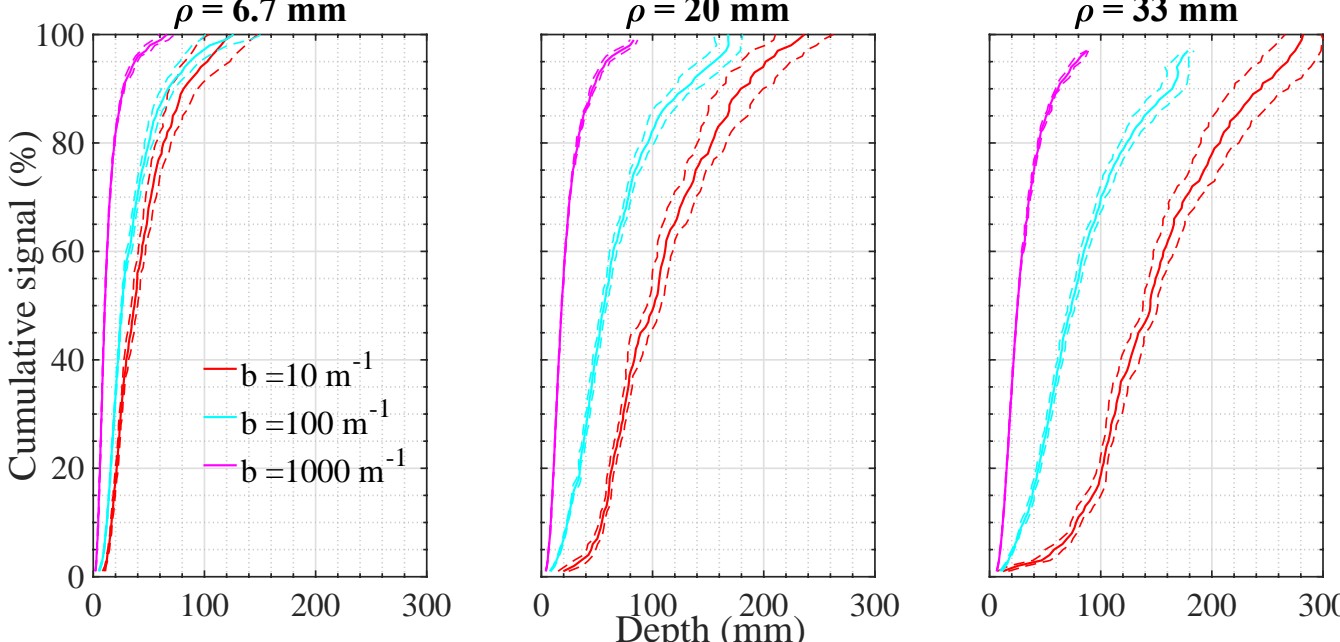

**Figure 6: Estimation of the depth of signal origin for our probe geometry simulated by Monte Carlo. The cumulative signal $R(z,r)/R_{zmax}(r)$ vs absorptive plate depth $z$ at different lateral distances $\rho$ was evaluated for three $b$ typical of sea ice, $a$=0.1m$^{-1}$ and g=0.94. We ran 40 simulations of 10$^6$ photons for every $z$ when $b$=10 m$^{-1}$ and 100 m$^{-1}$ and 10 simulations of 10$^6$ photons for every $z$ when $b$=1000 m$^{-1}$. The standard deviation on simulations was used to evaluate uncertainties.**

No matter $\rho_i$ and $b$, $z_{95}$ was always significantly greater than our minimum criterion of 10 mm (see Sect. 3.4) suggesting the signal originates from sufficiently deep to encompass even large scattering inclusions. In highly scattering ice, fibres $\rho_{4...7}$ had a $z_{95}$ greater than our maximum criterion of 50 mm. It implies that only the fibres at small $\rho_i$ should be used when scanning

the thin surface scattering layer with the probe leaning horizontally. In that case, detecting fibres should be chosen in function





of the scattering layer depth. Since we did not measure surface scattering layer properties in the case of this study, we did not further consider this maximum criterion.

## 4.2 Validation

Figure 7 compares the mean $a$, $b'$ and $\gamma$ to the theoretical values for 4 concentrations of microspheres in distilled water as

reference media. $a$ and $b'$ theoretical values covered close to two orders of magnitude and were typical of sea ice. The Mie phase function was forward peaked as for sea ice. The $e$ on measured IOPs is defined by Eq. (11). Fibres 1 and 7 were taken out of the inversion because fibre at $\rho_1 = 2\ mm$ was very close to the source and never met the criterion where $0.5 \leq \rho b'$ needed to be in the N=2 regime (see Sect. 2.2.2) and fibre at $\rho_7 = 43\ mm$ had a calibration factor $c_7$ roughly 10 times greater than calibration factors $c_{1-6}$. Either the simulated reflectance $\bar{R}_{sim}$ was not correctly modeled at this distance or fibre 7 was

damaged.

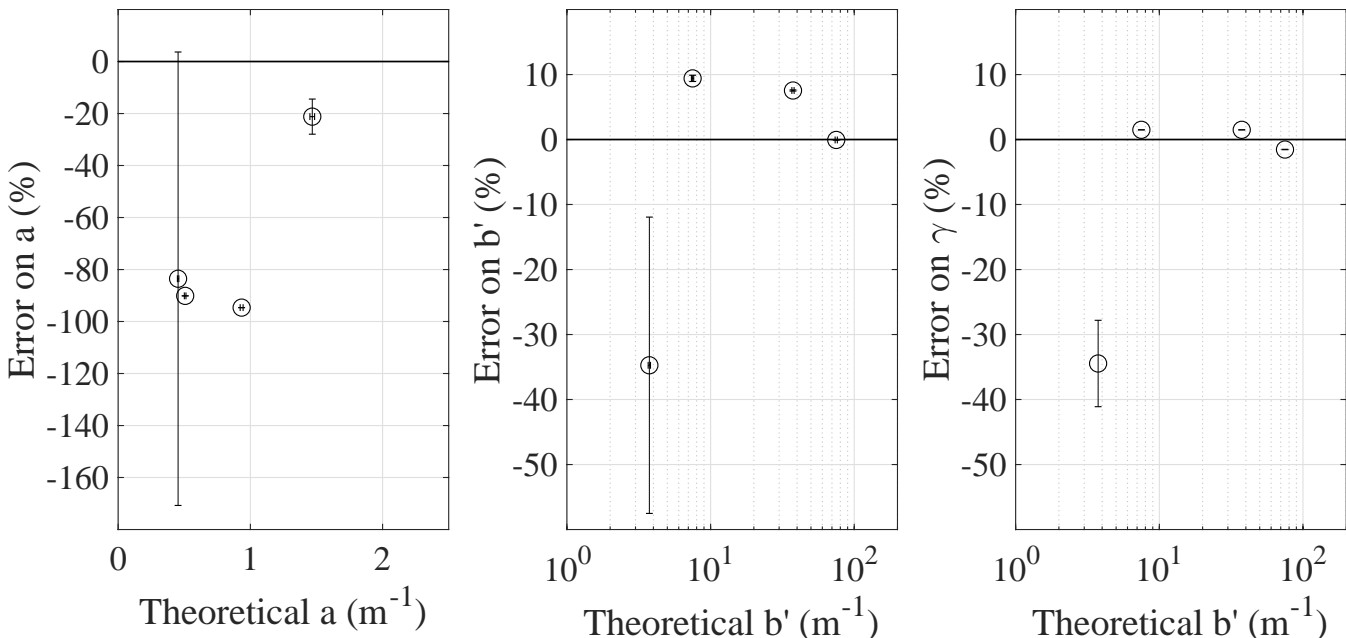

**Figure 7: Validation of IOPs measurements using reference media. The reference media were 4 solutions of polystyrene microspheres in distilled water. Microspheres concentrations were chosen for the theoretical $a$ and $b'$ ranges to cover sea ice typical values. Detecting fibres 1 and 7 were taken out for optimized results.**

Using fibres 2 to 6, we obtained $|e|$ between 21 % and 94 % for $a$, between 0.06 % and 35 % for $b'$ and between 1.5% and 34% for $\gamma$. These values are comparable to those obtained with classical instruments in marine optics (Leymarie et al., 2010). Standard deviations on $e$ were fairly low except at the lowest concentration where the inverted IOPs were very sensitive to signal variation. There, we obtained standard deviations of 87% on $a$, 23% on $b'$ and 7% on $\gamma$. Uncertainties on theoretically





calculated IOPs came from the uncertainty on the microspheres mean diameter. For the three parameters, the uncertainties
were less than 2 %.

**4.3 Physical properties of the sampled sea ice**

The first site was covered with 24 cm of snow. The ice thickness was 104 cm with a 2 cm freeboard. The second site was
uncovered (bare ice), therefore allowing more growth and thicker ice. The ice thickness was 135 cm with a 3 cm freeboard.
Ice cores were taken out roughly 5 metres away from optical measurement holes. Observations were made at the beginning of
the melt season when snow was starting to be slushy. A thin melt crust was present on snow at the snow-covered site.

Figure 8 a shows $T$ vertical profiles. At the snow-covered site, the coldest temperature was at the surface and was close to -
4°C. Temperature rose progressively and reached over -2°C at the bottom. At the bare ice site, temperature profile was c-
shaped. The temperature at the surface was over -2°C, the minimum was reached at 65 cm and was close to -4°C. The
temperature rose back to over -2°C at the bottom. The upper surface was warmer at the bare ice site because of the direct heat
transfer from air.

Figure 8 b shows $S$ vertical profiles. At the snow-covered site, bulk salinity profile was c-shaped. The bulk salinity at the
surface was 6.6 ‰. This value was high enough to suggest the uppermost boundary was formed of sea ice only. The minimum
was reached at 65 cm and was 3.4‰. Bulk salinity rose back to over 7.2 ‰ at the bottom. At the bare ice site, the uppermost
section had a bulk salinity of 2.7 ‰ which suggests brine drainage by gravity and melting. The bulk salinity stayed close to 5
‰ through the ice core and rose to 7.7 ‰ at the bottom section.



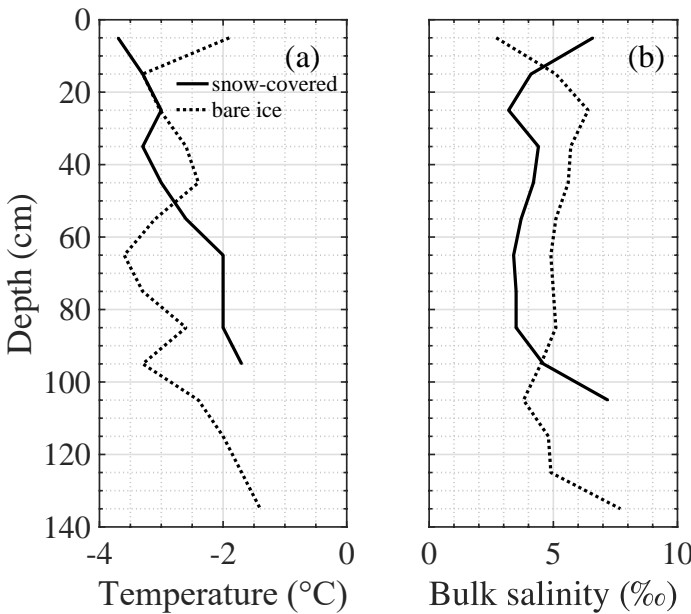

**Figure 8: Profiles of (a) $T$ and (b) $S$ measured on ice cores nearby optical measurement holes at both sites.**

## 4.4 Vertical profiles of $b'$ with fixed $a$ and $\gamma$

At both sites, vertical profiles of $b'$ were acquired with the probe in a two-inch auger hole with $a$ and $\gamma$ fixed. The motivation
to fix $a$ and $\gamma$ was to reduce their influence on inverted $b'$ ; we fixed $a$ to 0.22 m$^{-1}$ because $e$ was close to -100% on the range
corresponding to pure sea ice at $\lambda$=633 nm (see Figure 7). The value we chose corresponds to pure ice at $\lambda$= 633 nm (Picard
et al., 2016). Also, we fixed $\gamma$ to 1.98 when scanning sea ice and to 1.86 when scanning snow because $\gamma$ measurements in the
ice were highly noisy. These values were obtained assuming $g_1$ were 0.98 and 0.86 respectively and assuming the phase
function followed a Henyey-Greenstein distribution ($g_2 = g_1{}^2$). These values and distribution are commonly used to represent
$p(\theta)$ of sea ice in larger-scale radiative transfer calculation (Grenfell 1983). Finally, b$'$ measurements were not considered if
$R_{mes}$ and $\overline{R}_{sim}$ were off by more than 40% for either fibre 2 or 3. This criterion was the best we found to take out false inversion
results.

Measurements of $b'$ were acquired every 10 cm from the snow or ice surface until we reached the bottom (Figure 9).
Measurements were repeated at the same location with and without a tent covering the measurement hole to limit incident
sunlight. At the bare ice site, a profile with a 3 x 3 m tarpaulin fixed to the ground as a sunshade was also obtained. The same
profiles with and without subtraction of sunlight background $\Phi_{bg}$ from $R_{mes}$ were compared (see Eq. (8)).



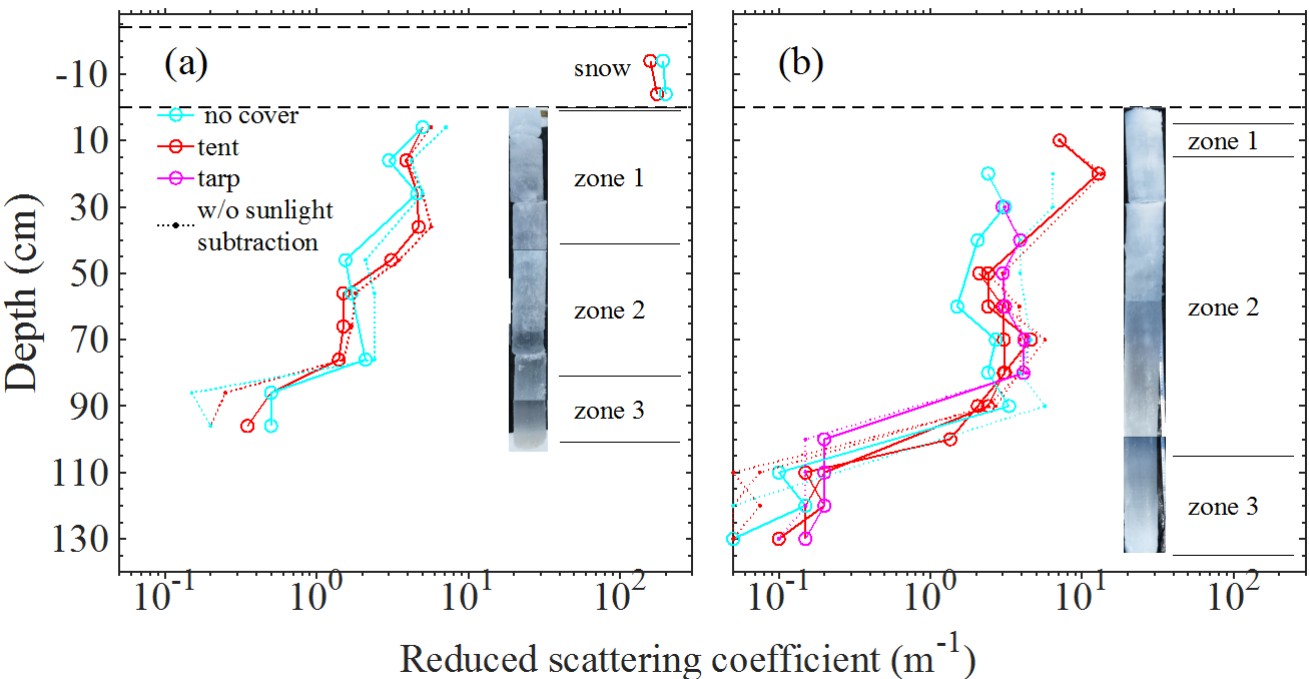

**Figure 9: Reduced scattering coefficient $b'$ vs depth measured actively in situ inside first-year Arctic interior sea ice. Measurements were taken (a) at a snow-covered site and (b) at a bare ice site. Pictures show ice cores extracted few meters away from the sampling holes. The profiles were separated by zones at depth where we observed significant changes in the value of $b'$. We compared measurements using different covers to shade from sunlight. We also compared with and without subtracting the residual sunlight background flux $\Phi_{bg}$.**

For both sampling sites, we observed a decrease in the value of $b'$ from the top to the bottom of the ice. We divided the profile into zones at depths where the $b'$ values changed significantly. Here, the uncertainties represent the standard deviation for every measurement within the given depth interval. At the snow-covered site, we observed four different zones. The average $b'$ for snow was $160\pm10$ m$^{-1}$, the average $b'$ for depths between 6 and 36 cm was $4.4\pm0.7$ m$^{-1}$, the average $b'$ for depths between 46 and 76 cm was $2.1\pm0.8$ m$^{-1}$ and the average $b'$ for depths between 86 and 96 cm was $0.46\pm0.07$ m$^{-1}$. At the bare ice site, we observed three different zones. The $b'$ value at a depth of 10 cm was 7.1 m$^{-1}$. The average $b'$ for depths between 20 and 100 cm was $2.8\pm0.8$ m$^{-1}$. For this zone, two measurements at the boundaries were discarded. One of those two was taken at a depth of 20 cm and had a value of 12.8 m$^{-1}$, which is much greater than the standard deviation. We believe the measurement was taken slightly closer to the surface and would have been more representative of the first zone of ice. The other measurement taken at a depth of 100 cm was 0.2 m$^{-1}$. We think the measurement was taken slightly deeper and therefore was included in the





third zone. The average $b'$ for depths between 110 and 130 cm was 0.15±0.05 m$^{-1}$. It is believed that the variability in the measurements was the consequence of heterogeneity in the ice morphology.

Ice core photographs from both sites showed a change in color from whitish to translucid, which was consistent with $b'$ vertical decay. At the snow-covered site, the transition from zone 1 to zone 2 at a depth of 46 cm was barely apparent on the picture. It corresponded to a drop of the mean $b'$ from 4.4±0.7 m$^{-1}$ to 2.1±0.8 m$^{-1}$. This was expected since the measurements of the two zones almost overlapped. The transition from zone 2 to zone 3 was easily distinguished on the picture. It corresponded to a drop of the mean $b'$ from 2.1±0.8 m$^{-1}$ to 0.46±0.07 m$^{-1}$. This was a 5-fold drop. However, the transition occurred 6 cm to 16

cm shallower on the picture. The ice cores were retrieved approximately 5 m away from the optical measurement sites. Therefore, the spatial variation in the ice structure might explain the imperfect depth consistency between $b'$ vertical profiles and the picture.

The three pictures stitched together were taken at different angles, and therefore sunlight reflection gave the false impression of a transition in the ice. This effect was more obvious at the bare ice site which made the comparison to the picture too

difficult.

## 5 Discussion

### 5.1 Sensitivity of $b'$ to $a$ and $\gamma$

The inverted $b'$ depend on the choice of fixed $a$ and $\gamma$ values. Thus, we analyzed the sensitivity of $b'$ to these two parameters for the profiles presented on Figure 9. Varying fixed $a$ from 0.01 to 0.5 m$^{-1}$ induced no significant variation on $b'$ profiles

except in the very low scattering ice of zone 3 where it induced variation of up to 50%. However, for every zone, varying fixed $\gamma$ from 0.8 to 1.98 induced very significant variations of up to an order of magnitude on inverted $b'$.

Our choice of fixed $\gamma$=1.98 is the value representing Henyey-Greenstein $p(\theta)$ with $g_1 = 0.98$ (thus $g_2 = 0.98^2$ ) (see Eq. (7)). This choice is commonly used for larger-scale radiative transfer in interior sea ice (Grenfell 1983), where only $g_1$ is relevant. Better insight into the in situ $p(\theta)$ of sea ice would be needed to say if the Henyey-Greenstein distribution represents

realistically its moments $g_{2+}$. Indeed, the knowledge of $g_2$ range in sea ice could be used to restrain $\gamma$ to a smaller, more plausible range, reducing the uncertainty on $b'$.

But even then, the sensitivity to $g_{3+}$ in low scattering ice would still remain a source of uncertainty. Since inverting on three parameters ($a$, $b'$ and $\gamma$) is already challenging, it is inconceivable to add higher phase function similarity parameters to account for the dependency to $g_{3+}$ in the inversion. Not to mention that barely more useful structural information would be

carried by these higher similarity parameters. If $g_{3+}$ values are relatively stable in sea ice, one could simply find a $p(\theta)$





modeling $g_{3+}$ of sea ice accurately for calculation of the lookup table. If $g_{3+}$ values are variable in sea ice, one could replace $\gamma$ by a phase function similarity parameter, like the $\sigma$ parameter introduced by Bodenschatz et al. (2016), which combines the contribution of every phase function moments $g_n$.

## 5.2 Comparison to previous measurement methods

Figure 10 compares the vertical average of $b'$ obtained for both sampling sites to the vertical averages of $b'$ measured in the past on polar interior sea ice using different methods (see Sect 2.2). Trodahl et al. (1987) and Mobley et al. (1998) reported a single value, while Light et al. (2008) and Light et al. (2015), reported ranges that represent a seasonal evolution. For Ehn et al. (2008a), the range represents different ice types. Average $b'$ for the different zones are also shown to illustrate the variability within interior sea ice. The error bars on our measurements represent the standard deviation for all $b'$ measurements in the

given depth interval.

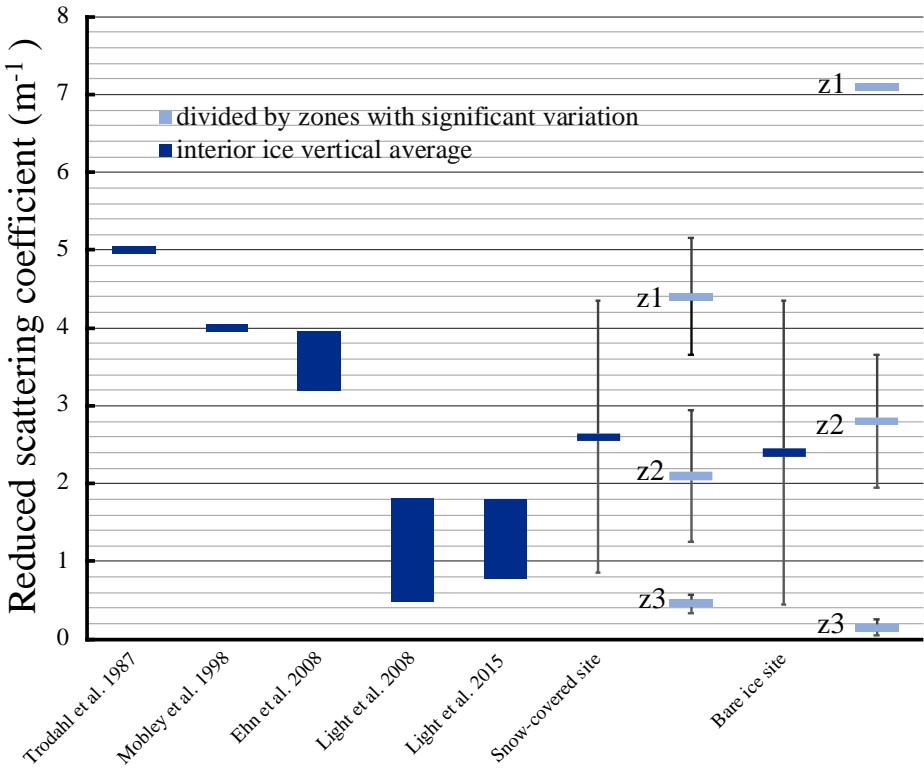

**Figure 10: Comparison of the vertical averages of $b'$ measured in situ at both sampling sites to vertical averages of $b'$ published over the past for polar interior sea ice using various methods. For our measurements, the average reduced scattering coefficients $b'$ is also divided by zones. Zones are separated at depths where we observed significant changes in scattering properties. The**
**error bars represent the standard deviation for the given depth interval.**





For our field work, the vertical average of $b'$ was 2.6±1.7 m$^{-1}$ for the snow-covered site and was 2.4±1.9 m$^{-1}$ for the bare ice site. These values are comparable to vertical averages of $b'$ measured by other authors. They are smaller than Trodahl et al. (1987), Mobley et al. (1998) and Ehn et al. (2008a), but greater than Light et al. (2008) and Light et al. (2015).

Our method was sufficiently resolved to observe the transitions in the optical properties of the interior ice in situ, demonstrating a strong vertical variability. Indeed, the standard deviations on the vertical average of $b'$ represented 65% and 80% of the mean. Furthermore, for both samples, average $b'$ of the uppermost zone was an order of magnitude greater than average $b'$ of the bottommost zone. This high vertical variability in the properties of interior sea ice highlights the necessity of a vertically resolved in situ technique like spatially resolved diffuse reflectance, allowing to divide the vertical profile into smaller, more

appearance, temperature, salinity,porosity, inclusion size and shape distributions) particular to the given zone.

## 5.3 Comparison to ice core images

Table 2 shows $b'$ measurements together with ice core pictures for the three zones we identified at the snow-covered site. Our pictures focused on the central depth of the zones in order to avoid ambiguities. These were compared to the $b'$ measurements and pictures of drained ice and interior ice by Light et al. (2008). No optical measurements were taken the day the core was

photographed (July 17$^{th}$). Thus we define the upper and lower limits of b' to be the closest measurement dates (July 9$^{th}$ and 21$^{st}$).




| | $b'$ (m$^{-1}$) | Photograph |
|---|---|---|
| Light et al. 2008 | 3-10 | |
| | 1.8 | |
| Snow-covered site | 4.4±0.7 | |
| | 2.1±0.8 | |
| | 0.46±0.07 | |


**Table 2: Average $b'$ measurements together with ice core pictures for the three zones we identified at the snow-covered site compared to the $b'$ measurements and pictures of drained ice and interior ice adapted from Light et al. (2008).**

The drained ice sample of Light et al. (2008) was distinctively whiter than the samples of zones 1 and 2 of our ice core. $b'$
range associated to the drained ice sample is overlapping the range of zone 1. However, the value of b' associated to the drained ice core photograph is probably closer to 10 m$^{-1}$, being closer in time to the upper limit. The interior ice sample of Light et al. (2008) showed a similar appearance as the ice of zone 3, yet with more white heterogeneities. Again, this goes along with the $b'$ measurement being greater than our value. The comparison of $b'$ measurements and ice core pictures is consistent with Light et al. (2008) , which suggests our measurements to be in the right range.





## 5.4 Comparison to structural-optical model

Using Grenfell (1983) framework, Light et al. (2003a) and Light et al. (2004) established a structural-optical model to determine $b'$ of an ice sample in a laboratory for T starting from -30°C to 0°C. (see Sect. 2.2). This $b'$ vs $T$ relationship was established for a salinity S of 4.7 ‰ which is close to what we measured. We compared our $T$ profiles (see Figure 8a) to their relationship in order to retrieve $b'$. At both sites, the estimated vertical average of $b'$ was 8.8±0.1 m$^{-1}$. It is to keep in mind that keeping S constant to 4.7 ‰ reduced the variability of estimated $b'$.

These values were roughly 3.5 times greater than what we measured in situ with the reflectance probe. We suppose this is because the laboratory model was corrected for brine channels drainage occurring when extracting the core from the ice. Channel drainage augments refractive index difference between the channel and the ice which augments scattering efficiency and affects the phase function. Change of gas volume and inclusions fusion might also affect scattering differently in the laboratory.

The extension of the structural-optical model to obtain in situ estimation of the IOPs is a difficult task. In particular because estimation of $b'$ requires knowledge of the distributions of sizes and shapes of the scattering inclusions in sea ice. Over the past, these distributions were estimated by means of microscope observations in a cold laboratory (Light et al., 2003a). Combining vertically resolved in situ $b'$ measurements to laboratory structure observations could potentially bridge that difficulty. Indeed, the structural-optical framework could be tuned to meet *in situ* $b'$ measurements the same way it is tuned to meet laboratory active $b'$ measurements in Light et al. (2004). For example, brine channel drainage, scattering by hydrohalite salts, the brine channels merging and air bubbles merging and escaping could be adjusted to meet in situ conditions. The estimation of in situ $b'$ based on the temperature, bulk salinity and size and shape distributions of the scattering inclusions of sea ice would be an interesting breakthrough. Indeed, it opens the door to the empirical estimation of in situ $b'$ based on sea ice growth conditions.

## 5.5 Error analysis

### 5.5.1 Estimation of error in sea ice based on validation

To estimate the error made on $b'$ when measuring inside sea ice, we compared in situ measurements to the instrument validation measurements made on microspheres solutions. The $b'$ validation points closest to our given $b'$ measurement inside ice established the boundaries of the corresponding error. Table 3 summarizes the instrument validation error $e$ attributed to the average of $b'$ of the different ice zones. For this analysis to make sense, we assumed that $e$ was the lowest at its calibration point ($b'$=75±1 m$^{-1}$) and that $e$ diverged as we moved away from this point.





| Site | Zone | Depth (cm) | Mean $b'$ (m$^{-1}$) | Instrument validation error (%) |
|------|------|------------|----------------------|--------------------------------|
| Snow-covered | Snow | -24 to 0 | 160±10 | - |
| | 1 | 6 to 36 | 4.4±0.7 | 9.4±0.5 to-35 ±20 |
| | 2 | 46 to 76 | 2.1±0.8 | <-35±20 |
| | 3 | 86 to 96 | 0.46±0.07 | <-35 ±20 |
| Bare ice | 1 | 10 | 7.1 | 9.4±0.5 to -35 ±20 |
| | 2 | 20 to 100 | 2.8±0.8 | <-35±20 |
| | 3 | 110 to 130 | 0.15±0.05 | <-35±20 |

**Table 3: Summary of mean $b'$ for the different ice and snow zones of both sampling sites and the corresponding instrument validation error $e$ . Zones were separated at depth where we observed significant changes in $b'$. The corresponding $e$ was based on the validation with microspheres solutions.**

For both sites, zone 1 had a $e$ between -35% and 9.4 % and zones 2 and 3 had $e$ below -35%. Because the $b'$ values in zone 2 were close to the validation point at $b'=3.75\pm0.07$ m$^{-1}$, we assumed that e=-35% was a decent estimation. The $b'$ values in zone 3 were far from the closest validation point. Therefore, e on these values was probably greater than $-35\%$. Because scattering for this zone was very low, we assumed that its influence on radiative transfer was also low. We can therefore tolerate higher $e$ for this zone.

The $e$ attributed to the different ice zones might be overestimated especially because we measured in low scattering ice; During validation with microspheres solutions, the measurement of $b'$ was altered by the depth of the container (~15 cm). For low scattering medium (b=10 m$^{-1}$), roughly no signal was cut for the first measuring fibres, but up to 20% of the signal originated from deeper than the container at fibre 4 and 45% at fibre 6 (see Figure 6). Because $R_{mes}$ was cut, inverted $b'$ was underestimated compared to the theoretical value, which might explain higher error $e$.

The $e$ could also be high because IOPs measurement in low scattering media is intrinsically difficult. As mentioned in Sect. 2.2.2, we assumed in our model that a phase function with two moments was sufficient to model reflectance. However, in media with low scattering ice properties, we do not meet the minimum criterion on the optical distance $\rho b' > 0.5$ mandatory for this assumption to be true. Not considering the moments of the phase function greater than two is therefore likely to lead to the underestimation of $b'$ in the validation.

### 5.5.2 Additional errors induced when measuring inside sea ice

We must keep in mind that this estimation of the measurement error inside sea ice is by no means absolute as there are key differences between measurements inside sea ice and measurements on the microspheres solutions used for validation. While $a$ and $\gamma$ are free in the validation, they are fixed when measuring inside sea ice. As mentioned in Sect. 5.1, the inversion of $\gamma$ can have a significant effect on the inversion of $b'$.





Then, the sunlight background is not negligible and represents an additional source of error when measuring inside sea ice. Field trials have shown that it can have the same order of magnitude as the signal in the worst scenario and be $10^4$ times smaller in the best scenario. Even though the sunlight background radiant flux $\Phi_{bg}$ is measured on the way up after profiling and subtracted from $R_{mes}$, it left a noise on the measurements nevertheless. This is the reason why we obtained different $b'$ profiles

depending on the sun shading method (see Figure 9).

## 6 Conclusion

We adapted a method to measure vertically resolved in situ inherent optical properties (IOPs) of sea ice. Conceptually, the spatially resolved diffuse reflectance $R_{mes}(\rho)$ measured from the ice interface is compared to a Monte Carlo simulated lookup table. The inversion algorithm inverts the absorption coefficient $a$, the reduced scattering coefficient $b'$ and the phase function

parameter $\gamma$ of a constrained volume (~dm³). Monte Carlo simulations showed that the depth cumulating 95% of the signal $z_{95}$ is between 40±2 mm and 270±20 mm depending on the source-detector distance $\rho$ and on the ice scattering properties. Validation of the measurements with microspheres solutions showed that the magnitude of the instrument validation error $|e|$ was between 21 % and 94 % for $a$, between 0.07 % and 35 % for $b'$ and between 1.5% and 34% for $\gamma$. The $|e|$ on $b'$ measurements was evaluated over close to two orders of magnitude corresponding to values typical of low and medium

scattering sea ice.

We tested the probe on first-year Arctic interior sea ice at two study sites around Qikiqtarjuaq Island next to Baffin Bay in Nunavut, Canada from May 7 to May 10, 2019, at the very beginning of the melt season. In the light of validation results, we fixed a to 0.22 m$^{-1}$ and γ to 1.98 and focused on the dominant and easier to retrieve $b'$. We measured every 10 cm sideward on the edge of an auger hole drilled through the ice. At the snow-covered site, we obtained $b'$ of 4.4±0.7 m$^{-1}$ for the uppermost

zone of interior ice and $b'$ of 0.46±0.07 m$^{-1}$ for the bottommost zone. At the bare ice site, we obtained a single $b'$ measurement of 7.1 m$^{-1}$ for the uppermost zone of interior ice and $b'$ of 0.15±0.05 m$^{-1}$ for the bottommost zone. These $b'$ measurements are sensitive to the choice of $\gamma$, revealing the need for a better representation of the higher moments of the in situ phase function of sea ice. Our results emphasize the strong vertical variability of the scattering properties even within interior sea ice. These values are in the range of polar interior sea ice mean $b'$ measurements previously published with different methods by other

authors. We demonstrated that the magnitude of $b'$ was consistent with the appearance of the ice core at the snow-covered site.

We believe combining vertically resolved in situ $b'$ measurements to laboratory structure observations could help to bridge structural and optical knowledge of sea ice. Indeed, the structural-optical framework of Grenfell (1983) could be tuned to meet in situ $b'$ measurements the same way it was tuned to meet laboratory active $b'$ measurements in Light et al. (2004). Combining in situ and laboratory observations could open the door to empirical radiative transfer estimations based on sea ice growth

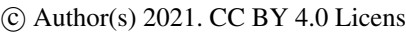



conditions. We believe that further developments of the spatially resolved diffuse reflectance method will lead toward a more widespread study and an improved comprehension of in situ IOPs of sea ice.

## Authors contributions

Pierre Marquet contributed to the original biomedical diffuse reflectance probe that inspired the sea ice IOPs probe. Pierre Marquet, Marcel Babin and Simon Lambert-Girard conceptualized the sea ice diffuse reflectance probe. Christophe Perron developed the inversion algorithm and the lookup table. Christophe Perron, Simon Lambert-Girard and Christian Katlein designed, built and collected data with the sea ice prototypes on the field and designed the microsphere validation experiment. Christophe Perron and Christian Katlein developed the probe operation software. Edouard Leymarie provided the software SimulO used for the optical simulations. Marcel Babin, Pierre Marquet and Edouard Leymarie provided mentorship throughout the whole development process. Louis-Philippe Guinard, Simon Lambert-Girard, Christian Katlein and Christophe Perron designed the ruggedized prototype after field trials and conducted validation tests. Marcel Babin directed the project. Pierre Marquet co-directed the project. Christophe Perron prepared the initial draft of the manuscript and prepared visualizations. All co-authors reviewed and edited the draft.

## Competing interests

The authors declare that they have no conflict of interest.

## Disclaimer

No Disclaim

## Acknowledgements

We would like to thank Philippe Massicotte who helped to build the lookup table and to run the simulations on the supercomputers of Compute Canada. Guislain Bécu and Jose Luis Lagunas helped to print the probe heads and whom provided technical support throughout the whole development process. Marie-Hélène Forget for her precious logistic support during field deployment in Arctic and throughout the whole process. Philippe DeTilleux for helping in results validation and helping in developing the microspheres experiment. Jean-Marie Trudeau and Éric Barucha for counseling and providing instruments. Frederic Maes for cleaving the optical fibres. Daniel Coté for advice on instrument development and optical simulations. Flavienne Bruyant for her logistic help during laboratory tests. Félix Levesque-Desrosiers, Yasmine Alikacem and Raphael Larouche for the insightful discussions.





Finally, the research project was supported by the Canada Excellence Research Chair on remote sensing of Canada's new Arctic frontier, Discovery grant #RGPIN-2020-06384 to Marcel Babin, the SMAART programme funded by the Collaborative Research and Training Experience Program of the Natural Sciences and Engineering Research Council of Canada and the Sentinel North program of Université Laval, made possible, in part, thanks to funding from the Canada First Research
Excellence Fund.

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
