# Peer review of "Development of a diffuse reflectance probe for in situ measurement of inherent optical properties in sea ice"

_The Cryosphere, 2021_

## Referee Comment (RC1)

**Review:**

**Development of a diffuse reflectance probe for in situ measurement
of inherent optical properties in sea ice**

Christophe Perron1,4, Christian Katlein1,2, Simon Lambert-Girard1, Edouard Leymarie3, Louis-Philippe Guinard1,4, Pierre Marquet4,5, Marcel Babin1
1Takuvik Joint International Laboratory, Laval University (Canada)-CNRS (F 5 rance), Québec city, G1V 0A6, Canada
2Alfred-Wegener-Institut Helmholtz-Zentrum für Polar- und Meeresforschung, Bremerhaven, 27570, Germany
3Laboratoire d'Océanographie de Villefranche-sur-Mer, Villefranche-sur-Mer, 06230, France
10 4CERVO Brain Research Centre, Laval University, Québec city, G1J 2G3, Canada
5Centre d'optique, photonique et laser, Laval University, Québec city, G1V 0A6, Canada
*Correspondence to*: Christophe Perron (christophe.perron@takuvik.ulaval.ca)

**General comments**

This is a well written paper that is based on a very substantial and impressive body of work by the authors. The authors developed and extensively tested a new diffuse reflectance for sea-ice. They carried out a detailed ice scattering model analysis using a complete Monte-Carlo code and experimentally validated its functioning and calibration using micro-spheres suspensions and a Mie code scattering phase function model. Their probe was then used in-situ to analyze the inherent properties of sea-ice as a function of depth from the surface. The results of this work and the resulting measurement techniques are extremely relevant and could be used as a starting point to ultimately obtain functional models of sea-ice generation and loss in natural environments. The probe and the signal analysis techniques give a glimpse of the possible performance and environment monitoring accuracy improvements obtainable from their use. This information will be extremely useful to other researchers in the field. For the reasons above I recommend publication of this paper. There are however several developments in the paper which, at the author's discretion, could, in my opinion, be improved before publication. I have noted those more serious problems and along with minor deficiencies/improvements in my comments below.

**Specific comments**

Suggestions for improvements

Line 95 and following:
I have had a problem in following the original theoretical introduction because of missing terms in the discussion. I would add the definition of the moments immediately after equation 2.
Where $g_n$

$$g_n = 2\pi \int_0^\pi P_n(\cos\theta)\, p(\theta) \sin\theta\, d\theta$$

$P_n(\cos\theta)$ are the Legendre polynomials and $\theta$ denotes the angle between incident photon direction and photon direction after scattering. The first three Legendre polynomials which we will use are:
$$P_0(\cos\theta) = 1$$

$$P_1(\cos\theta) = \cos\theta$$

$$P_2(\cos\theta) = \frac{1}{2}(3\cos^2\theta - 1)$$

Equation 3 then becomes

$$g_1 = 2\pi \int_0^\pi p(\theta)\cos\theta\sin\theta\,d\theta$$

Note that the solid angle element of integration $\sin\theta\,d\theta$ has been shifted to the end of the integral to separate it from the function being integrated over the solid angle to keep the physics underlying the equation clearer.

Line 145:
All the subsequent higher moments after the second moment of the modified phase function are simply

$$g_n = \beta g_{HG}^n \ \text{ for } n > 2$$

Since the integral of $(1-\beta)\cos^2\theta\, P_n(\cos\theta)$ term is identically zero due to the orthogonality of the Legendre functions for any $n \neq 2$. This fact should be mentioned since at the end of the paper there is some discussion of the importance of the higher moments. The conclusion above implies that those higher moments and any of their ratios are basically controlled by the $g_{HG}$ parameter which considerably limits any flexibility to model more complex situations as the behavior of the solution as a function of $g_{HG}$ is already fully accounted for in the current model.

Line 232:
The fact that the laser emitter cone does not have the same angular range as the NA of the fiber in ice is an indication that the fiber does not completely scramble the laser input and significant traces of the fiber input conditions remain at the fiber exit. This is not surprising for such a short fiber with a single bend. It's a known problem in diode pumped lasers. This however implies that care must be taken not to disturb the fiber by moving it after the measurement of the output beam is done. Ultimately this problem can be corrected by using a longer fiber and winding it on a mandrill or around the cavity of the probe. However, given the minimum bend radius of the fiber, you may not have enough room in the probe in which case I would recommend making sure the fiber is fixed in place by a holder or support.

Line 395
The effect of the container wall of the theoretical values of reflectivity for the polystyrene sphere suspensions should be expanded as they could be a substantial portion of the errors which seem to occur predominantly at the low values of absorption and scattering. The authors mention this in the discussion and conclusions but it should be further addressed at this point to at least indicate clearly what results are significantly subject to the wall influence.

Line 610

As a suggestion for future work and to start bridging the gap between structural and optical knowledge the researchers could use the vast and valuable simulation data base to reevaluate the behavior of the absolute and relative reflectivity as a function of different non-dimensional parameterizations to identify

the significant correlations. Two parameters come to mind immediately, the backscatter coefficient and the absorption over b' .

$$\frac{b_b(\beta, g)}{b} = 2\pi \int_{\pi/2}^{\pi} p(\beta, g, \theta) \sin\theta \, d\theta = \frac{\beta}{2 \, g}\left[\frac{(1-g^2)}{\sqrt{(1+g^2)}} - (1-g)\right] + \frac{(1-\beta)}{2}$$

The second parameters of interest could be the asymptotic value of the mean cosine (first moment). Piskozub and McKee (see attached reference) have shown that the limit of the first moment of the radiation distribution after many collisions is given by:

$$g_\infty = \frac{g(1-\omega)}{(1-g\,\omega)}$$

$g$ is the first moment of the scattering function for the first collision and $g_\infty$ is the resulting radiation distribution after a large number of scattering collisions. $\omega$ is the albedo

$$\omega = \frac{1}{\left(\frac{a}{b}\right) + 1} = \frac{1}{\left(\frac{a}{b'}\right)(1-g) + 1}$$

$$g_\infty = \frac{g\left(\frac{a}{b'}\right)}{\left[1 + \left(\frac{a}{b'}\right)\right]}$$

This indicates that the parameter $\frac{a}{b'}$ is also a candidate for which the correlations should be looked at

Finally the simple scaling against $b'z$ could be used to analyze the correspondence of the computed reflectivity at the different detectors. Detectors with identical $b'L$ where $L$ is the distance between the source and the detector should have the same response if $\frac{a}{b'}$ and $\frac{b_b}{b}$ are the same.

Reference [1]
"Effective scattering phase functions for the multiple scattering regime"
Jacek Piskozub and David McKee Optics ExpressVol. 19,Issue 5,pp. 4786-4794
https://doi.org/10.1364/OE.19.004786

**Technical comments**

1) Line 18 "by optical fiber" should be replaced by: "by an optical fiber"
2) Line 19 "receiving fibres" should be replaced by: "receiving fibers"
3) Line 21 "allowing to analyze" should be replaced by: "allowing the analysis of"
4) Line 29 "dependent of" should be replaced by: "dependent on"
5) Line 30"this novel developed probe" should be replaced by: "this newly developed probe"
6) Line 185 "assuring them" should be replaced by: "which assured them"
7) Line 191"of optical properties" should be replaced by:"of the optical properties"

8) Line 206"concise volume" should be replaced by:"compact volume"
9) Line 227"splitted" should be replaced by"split"
10) Line 268 "thereof" should be replaced by "therefore"
11) Line 274"inverse" should be replaced by "the inverse"
12) Line 298''detecting fibres" should be replaced by "the detecting fibers"
13) Line 331 "uncalibrated" should be replaced by "the uncalibrated"
14) Line 390"in function" should be replaced by "as a function"
15) Line 475"induced variation" should be replaced by "induced variations"
16) Line 536"Using Grenfell" should be replaced by "Using the Grenfell"
17) Line 543"refractive index" should be replaced by "the refractive index"
18) Line 586" Then" should be replaced by "Also"
19) Line 592"adapted" should probably be replaced by "developed and validated"
20) Line 610 "ice core" should be replaced by "ice cores"
21) Line 616"a more widespread study" could be replaced by "more widespread and wide ranging studies"

---

## Author Comment (AC1)

Color Code:

Normal= Reviewer comment
Green = Reviewer comment accepted
Yellow = Reviewer comment discussed
*Italics= Authors' response to the reviewer's comment*

**General comments**

This is a well written paper that is based on a very substantial and impressive body of work by the authors. The authors developed and extensively tested a new diffuse reflectance for sea-ice. They carried out a detailed ice scattering model analysis using a complete Monte-Carlo code and experimentally validated its functioning and calibration using micro-spheres suspensions and a Mie code scattering phase function model. Their probe was then used in-situ to analyze the inherent properties of sea-ice as a function of depth from the surface. The results of this work and the resulting measurement techniques are extremely relevant and could be used as a starting point to ultimately obtain functional models of sea-ice generation and loss in natural environments. The probe and the signal analysis techniques give a glimpse of the possible performance and environment monitoring accuracy improvements obtainable from their use. This information will be extremely useful to other researchers in the field. For the reasons above I recommend publication of this paper. There are however several developments in the paper which, at the author's discretion, could, in my opinion, be improved before publication. I have noted those more serious problems and along with minor deficiencies/improvements in my comments below.

*We thank the reviewer for its detailed and insightful response. The relevance of the comments regarding the structure of the theory will greatly contribute to the precision and understanding of the manuscript. Also, its propositions regarding future work are very good, smart and detailed and will certainly lead to interesting ideas.*

**Specific comments**

Suggestions for improvements

Line 95 and following: I have had a problem in following the original theoretical introduction because of missing terms in the discussion. I would add the definition of the moments immediately after equation 2. Where

$$g_n = 2\pi \int_{\theta=0}^{\pi} P_n(cos\ \theta) p(\theta)\ sin\ \theta d\ \theta$$

are the Legendre polynomials and θ denotes the angle between incident photon direction and photon direction after scattering.

The first three Legendre polynomials which we will use are:

$$P_0(\cos\theta) = 1$$

$$P_1(\cos\theta) = \cos\theta$$

$$P_2(\cos\theta) = \frac{1}{2}(3\cos^2\theta - 1)$$

Equation 32 then becomes

$$g_1 = g = 2\pi \int_{\theta=0}^{\pi} p(\theta)\cos\theta \sin\theta \, d\theta$$

Note that the solid angle element of integration has been shifted to the end of the integral to separate it from the function being integrated over the solid angle to keep the physics underlying the equation clearer.

Line 145: All the subsequent higher moments after the second moment of the modified phase function are simply

$$g_n = \beta g_{HG}^n \qquad \text{for } n > 2$$

Since the integral of term is identically zero due to the orthogonality of the Legendre functions for any . This fact should be mentioned since at the end of the paper there is some discussion of the importance of the higher moments. The conclusion above implies that those higher moments and any of their ratios are basically controlled by the parameter which considerably limits any flexibility to model more complex situations as the behavior of the solution as a function of is already fully accounted for in the current model.

Line 232: The fact that the laser emitter cone does not have the same angular range as the NA of the fiber in ice is an indication that the fiber does not completely scramble the laser input and significant traces of the fiber input conditions remain at the fiber exit. This is not surprising for such a short fiber with a single bend. It's a known problem in diode pumped lasers. This however implies that care must be taken not to disturb the fiber by moving it after the measurement of the output beam is done. Ultimately this problem can be corrected by using a longer fiber and winding it on a mandrill or around the cavity of the probe. However, given the minimum bend radius of the fiber, you may not have enough room in the probe in which case I would recommend making sure the fiber is fixed in place by a holder or support.

*Indeed, we also believe that the laser input is not completely scrambled when coming out of the fibre. We observed spiking when looking at the reflected spot coming out of the source fibre. For future work, maybe we could bend the fibre on a mandrel on top of the pole and verify the stability of the laser power.*

*When using the probe on the field, the user always held the pole in position for the 30 seconds interval between reference measurement and the last detecting fibre measurement. Neither the probe head nor the fibre bundle were moving during this time interval.*

*A probe holder is also used on the currently upgraded version to limit movements.*

Line 395 The effect of the container wall of the theoretical values of reflectivity for the polystyrene sphere suspensions should be expanded as they could be a substantial portion of the errors which seem to occur predominantly at the low values of absorption and scattering. The authors mention this in the discussion and conclusions but it should be further addressed at this point to at least indicate clearly what results are significantly subject to the wall influence.

*We agree that the effect of the container walls should be mentioned earlier in the manuscript as it probably accounts for an important part of the error in the validation with microspheres solutions. Indeed, the fact that the error is getting greater as b' diminishes correlates with the depth of signal origin increasing, and eventually getting greater than the depth of the container, as b' diminishes . We will elaborate this effect and clearly specify which calibration points are subject to this error in section 4.2-validation with microspheres based on our simulation of the depth of signal origin shown on figure 6.*

Line 610 As a suggestion for future work and to start bridging the gap between structural and optical knowledge the researchers could use the vast and valuable simulation data base to reevaluate the behavior of the absolute and relative reflectivity as a function of different non-dimensional parameterizations to identify the significant correlations. Two parameters come to mind immediately, the backscatter coefficient and the absorption over b'.

$$\frac{b_b(\beta, g)}{b} = 2\pi \int_{\pi/2}^{\pi} p(\beta, g, \theta) sin\theta d\theta = \frac{\beta}{2g} \left[ \frac{(1-g^2)}{\sqrt{(1+g^2)}} - (1-g) \right] + \frac{(1-\beta)}{2}$$

 The second parameters of interest could be the asymptotic value of the mean cosine (first moment). Piskozub and McKee (see attached reference) have shown that the limit of the first moment of the radiation distribution after many collisions is given by:

$$g_\infty = \frac{g(1-\omega)}{(1-g\omega)}$$

is the first moment of the scattering function for the first collision and is the resulting radiation distribution after a large number of scattering collisions. $\omega$ is the albedo

$$\omega = \frac{1}{\left(\frac{a}{b}\right) + 1} = \frac{1}{\left(\frac{a}{b'}\right)(1-g) + 1}$$

$$g_\infty = \frac{g\left(\frac{a}{b'}\right)}{\left(1 - \frac{a}{b'}\right)}$$

This indicates that the parameter is $\frac{a}{b'}$ also a candidate for which the correlations should be looked at.

*These suggestions are very interesting and will certainly lead to promising future work .*

*-For the first equation, it seems that $b_b$ (or $b_b$ (1-g)?) could be inverted from spatially resolved diffuse reflectance based on our current model. It would be interesting to obtain b or b (1-g) by other means either 1) using a radiance profiler and performing an inversion or 2) providing an estimation of the size and shape distributions of the brine channel and bubbles. Using the equation, one could then provide an estimation of the relative contribution of β and g (therefore γ).*

*-For the second, third equation and fourth equation, maybe $g_\infty$ could be measured either using a goniometer or a radiance profiler in the ice. a/b' could theoretically be obtained with the probe, but $a$ measurements are very imprecise at the moment. Using $g_\infty$ and a/b' estimations with equation 4, we might also retrieve g.*

Finally, the simple scaling against b'z could be used to analyze the correspondence of the computed reflectivity at the different detectors. Detectors with identical where is the distance between the source and the detector should have the same response if $\frac{a}{b'}$ and $\frac{b_b}{b}$ are the same.

*As mentioned in section 2.1, even when the source-detector distance is scaled by the reduced scattering coefficient $b'\rho$ (or b'z), other factor will affect the Reflectance. The geometry (mainly the acceptance angle of the fibre) will affect R. Maybe this first factor could be corrected normalizing R by the solid angle. The refractive index would also have an impact. Then, as mentioned in section 2.2.2, in N=2 regime, R will also be affected by γ.*

*Unfortunately, to our knowledge, no reflectance table provides R in function of acceptance angle and γ.*

Reference [1] "Effective scattering phase functions for the multiple scattering regime" Jacek Piskozub and David McKee Optics ExpressVol. 19,Issue 5,pp. 4786-4794 https://doi.org/10.1364/OE.19.004786

**Technical comments**

1) Line 18 "by optical fiber" should be replaced by: "by an optical fiber"
2) Line 19 "receiving fibres" should be replaced by: "receiving fibers"
   *The formulation fibre(s) was preferred because The Cryosphere Journal recommend to use UK English. This formulation is also used in Canadian English, the country of origin or the work country of most authors.*

3) Line 21 "allowing to analyze" should be replaced by: "allowing the analysis of"
4) Line 29 "dependent of" should be replaced by: "dependent on"
5) Line 30"this novel developed probe" should be replaced by: "this newly developed probe"
6) Line 185 "assuring them" should be replaced by: "which assured them"
7) Line 191"of optical properties" should be replaced by:"of the optical properties"
8) Line 206 "concise volume" should be replaced by "compact volume"
*Indeed, compact is more appropriate than concise. However, searching the definition of the word, the author meant "defined volume", meaning we can estimate the size of the volume.*
9) Line 227"splitted" should be replaced by"split"
10) Line 268 "thereof" should be replaced by "therefore"
11) Line 274"inverse" should be replaced by "the inverse"
12) Line 298"detecting fibres" should be replaced by "the detecting fibers"
13) Line 331 "uncalibrated" should be replaced by "the uncalibrated"
14) Line 390"in function" should be replaced by "as a function"
15) Line 475"induced variation" should be replaced by "induced variations"
16) Line 536"Using Grenfell" should be replaced by "Using the Grenfell"
17) Line 543"refractive index" should be replaced by "the refractive index"
18) Line 586" Then" should be replaced by "Also"
19) Line 592"adapted" should probably be replaced by "developed and validated"
20) Line 610 "ice core" should be replaced by "ice cores"
*Only one ice core was retrieved at the snow-covered site, so ice core should stay singular. We did not include the ice core of the bare ice site because the brightness makes it difficult to compare.*
21) Line 616 "a more widespread study" could be replaced by "more widespread and wide ranging studies"

---

## Author Comment (AC2)

**Final response to RC2- Anonymous Referee #2**

Code:
Green = Reviewer comment accepted
Yellow = Reviewer comment discussed
*Italics= Authors' response to the reviewer's comment*

**General comments**

This manuscript describes the development and testing of a novel instrument for direct measurement of the scattering coefficient in the interior of sea ice. Sea ice is a strongly multiply forward-scattering domain so direct measurements of the inherent optical properties have been challenging. This instrument uses an active optical test to acquire reflectance data used to interpret the spatial distribution of scattered light in a relatively small volume. A forward radiative transfer model is run for a wide range of scattering coefficients to generate a look-up table to which the observed reflectance pattern is compared. Results indicate that inferred scattering coefficients fall into the range of expected values.

The probe itself appears to have significant promise for investigation of the optical properties of sea ice. The manuscript describing the probe is comprehensive and does a good job of outlining the theoretical basis for the probe, its design, validation, and an example data set. The figures are clear and appropriate (one minor comment on Fig. 1, below). I have no substantial concerns about this manuscript and recommend it for publication. I was a bit surprised that the field tests did not include more information about the IOPs of the ice near its upper surface. Seems this is where this instrument could really shine, but it sounds as though there may be some technical issues to work through before the instrument can be used to interpret scattering through the entire column.

The remainder of my comments are minor and address the clarity of the language. There are numerous instances where the language is a bit imprecise, so obscures the intended meaning. I've attempted to point these out below. Otherwise, the presentation does a good job of motivating and explaining the hardware, results, and issues associated with data interpretation.

*We want to thank the reviewer for its precious contribution regarding the precision and clarity of the manuscript and for its corrections regarding some key elements of the cited litterature.*

**Specific comments**

19 – 22: sentence beginning "Comparison to a Monte Carlo." This sentence implies that all three IOPs can be inferred, whereas in practice it appears that satisfactory inversions are accomplished by assuming a and gamma? Also, this sentence should be broken into two sentences.

*We will add :"Comparison to a Monte Carlo simulated lookup table allows, in theory, to retrieve the absorption coefficient, the reduced scattering coefficient and a phase function similarity parameter $\gamma$,"*

*And* "Fixing the absorption coefficient and $\gamma$, which proved difficult to measure, vertical profiles of the reduced scattering coefficient were obtained with decimeter resolution on first-year Arctic interior sea ice on Baffin Island in early spring 2019." (line 25)

22-23: Sentence beginning "Monte Carlo simulations..." needs to be rewritten for clarity 29: strongly dependent *on* gamma?

*Will be rewritten as :"The depth reached into the medium by detected photons was estimated using Monte Carlo simulations: The maximum depth reached by 95% of the detected photons was between 40±2 mm and 270±20 mm depending on the source-detector distance and on the ice scattering properties."*

30: "novel probe" delete "developed"; also "scattering in sea ice" not "into".
32: govern (not "are governing")

45: "the vertical distribution of IOPs"

48: "approximations"

51: instead of enlightenment, solar insolation or incident illumination

85: Does G also depend on the viewing direction of the receiving fibers (enclosed angle between direction of centers of source and detector fibers)?

*The direction of the viewing fiber affects the depth of signal origin as demonstrated by one of our colleagues. By intuition, we could presume that the direction of the viewing fiber effect on R is strongly dependent on the angular profile of the backscattered light: under angularly homogeneous backscattered light, the direction of the viewing fibre would not affect R. Under strictly upward backscattered light, we believe the reflectance would diminish proportionally to the foreshortening of the effective detection area. That is if we don't consider the effects of refraction.*

*Regarding the refractive effects, the effect of the viewing direction on R would depend on whether the fibre is held in the uppermost media or tilted downward to dip in the bottommost media. The various refractive effects would be different in those 2 situations.*

*We did not include the direction of the viewing angle as part of G, because its effect is not well documented and because the correlation to geometry is not as obvious as for the other parameters. We assume the reader will understand that, in our case, the fibres are strictly perpendicular to the probed medium as it is illustrated on fig.1,3 and 4.*

97-98: fewer moments required as number of scattering events in the optical path augments. Do you mean "optical path increases"? Rewrite for clarity.

==122-125: I think it likely that Grenfell & Hedrick (1983) had difficulty isolating single scattering and were probably measuring a domain somewhere between single scattering and diffusion regime.==

*From Grenfell and Hedrick (1983):*

*"optical thickness through the ice for the most opaque samples was estimated using extinction data from Perovich(1979) to be less than 0.0025 - excluding ice below the eutectic point. This was assumed to be sufficiently optically thin to give a reasonable representation of single scattering"*

*But indeed their sample were 10 mm thick while scattering mean length of interior sea ice is roughly 10 mm at the lower extreme. Meaning their measurements are potentially biased by multiple scattering in the sample.*

*Because of the ambiguity, we decided to take off the sentence.*

128: Please check this reference.

*Corrected to Van de Hulst and Christoffel 1980*

134 (paragraph beginning): Is "N" defined? Is it the same as "n"? It is not clear exactly what is being evaluated here. What is meant by "set free"?

*n corresponds to the order of the Legendre polynomials moments $g_n$, while N represents the regime, meaning the number of free moments $g_n$ needed to correctly characterize the phase function. For example, the Henyey-Greenstein phase function has infinite moments $g_n$ described by $g_n = g_1^n$, but only one of them is a free moment ($g_1$). It therefore lies in the N=1 regime.*

*We will add the definition of N to the manuscript and precise what is meant by set free.*

==149: Please provide a reference for precipitated salt crystals that are smaller than the wavelength and thus serving as Rayleigh scatterers.==

*We realised that smallest known ice crystals are roughly 1 micron in size (according to Light 1995), which is roughly 1.5 times the wavelength. Therefore, Rayleigh scattering probably does not play an important role when it comes to salt crystals.*

*However, Rayleigh scattering, though not documented in sea ice litterature, could be caused by nanometric scale dislocations in the ice matrix, dissolved NaCl and insoluble dust particles (Price and Bergström, 1997). We will modify the text to correct the potential Rayleigh scatterers. At the same time, we will add the mathematical equation stating that for anisotropic scattering: $g_2 \leq g_1$ and for Rayleigh scattering $g_1=0$ and $g_2=0.1$ in order to be more precise.*

*P. B. Price & L. Bergström (1997) Enhanced Rayleigh scattering as a signature of nanoscale defects in highly transparent solids, Philosophical Magazine A, 75:5, 1383-1390, DOI: 10.1080/01418619708209861*

158 optically dense?: "impenetrable"? *We meant solid.*

169: Light et al Monte Carlo model uses reciprocity to solve the RT equation, but is not truly an inverse model.

Figure 1: would it be helpful to show an arrow going from "Filter & photodiode" to "Computer" to show that the measured light is compared with MC simulations?

*Indeed, the arrow were already included, but some element of the figure disappeared when converted to .pdf. We will make sure to correct this technical issue before submission.*

221: bandpass filtering at 633 nm designed to reject sunlight, but there is plenty of sunlight in the ice at this wavelength? Maybe just say "reject sunlight at extraneous wavelengths"?

307, 311, 390: horizontally? Not clear what this means?

*We will add this clarification" leaned horizontally (meaning the fibers are looking downward)"*

329 – 331: this last sentence could be omitted
416: lowest (not coldest) temperature
544 inclusions "fusion"? Maybe merging inclusions?